# Engineering of ultraID, a compact and hyperactive enzyme for proximity-dependent biotinylation in living cells

Lea Kubitz [1,8], Sebastian Bitsch[2,8], Xiyan Zhao[1,8], Kerstin Schmitt [3], Lukas Deweid[2,5], Amélie Roehrig[1,6], Elisa Cappio Barazzone[1,7], Oliver Valerius [3], Harald Kolmar[2] & Julien Béthune [4✉]

Proximity-dependent biotinylation (PDB) combined with mass spectrometry analysis has established itself as a key technology to study protein-protein interactions in living cells. A widespread approach, BioID, uses an abortive variant of the *E. coli* BirA biotin protein ligase, a quite bulky enzyme with slow labeling kinetics. To improve PDB versatility and speed, various enzymes have been developed by different approaches. Here we present a small-size engineered enzyme: ultraID. We show its practical use to probe the interactome of Argonaute-2 after a 10 min labeling pulse and expression at physiological levels. Moreover, using ultraID, we provide a membrane-associated interactome of coatomer, the coat protein complex of COPI vesicles. To date, ultraID is the smallest and most efficient biotin ligase available for PDB and offers the possibility of investigating interactomes at a high temporal resolution.

[1] Heidelberg University Biochemistry Center, Heidelberg, Germany. [2] Institute for Organic Chemistry and Biochemistry, Technische Universität Darmstadt, Darmstadt, Germany. [3] Institute of Microbiology and Genetics, Göttingen Center for Molecular Biosciences (GZMB) and Service Unit LCMS Protein Analytics, Georg-August-University Göttingen, Göttingen, Germany. [4] Department of Biotechnology, Hamburg University of Applied Sciences, Hamburg, Germany. [5] Present address: Ferring Pharmaceuticals, Copenhagen, Denmark. [6] Present address: Inserm UMRS1138 – FunGeST team, Paris, France. [7] Present address: Department of Health Sciences and Technology, ETH Zürich, Zürich, Switzerland. [8] These authors contributed equally: Lea Kubitz, Sebastian Bitsch, Xiyan Zhao. ✉email: Julien.bethune@haw-hamburg.de

Proximity-dependent biotinylation (PDB) is a powerful technique to identify protein–protein interactions in living cells[1]. In the original approach BioID, the R118G abortive variant of the *Escherichia coli* protein biotin ligase (PBL) BirA, is fused to a protein of interest[2]. In cells, BioID uses ATP and biotin as substrates to produce biotinyl-AMP (bioAMP). By contrast to the wild-type (WT) enzyme, BioID does not retain bioAMP in its active site but releases it, leading to the non-enzymatic biotinylation of lysine residues on surrounding proteins within an estimated radius of about 10 nm (ref. [3]). Such labeled proteins can then be affinity purified on a streptavidin matrix and identified by mass spectrometry.

BioID is not only a valuable and widely used tool but also suffers from drawbacks. First, it is rather bulky (approx. 36 kDa), which sometimes hampers the proper localization of the corresponding fusion proteins[4]. Second, its kinetic of labeling is slow, necessitating hours of incubation with exogenous biotin[2,4]. Finally, as *E. coli*'s BirA is a class II biotin ligase, it has an N-terminal DNA-binding domain[5] that shows structural homology to linker histone H5[6] and that theoretically might lead to artefactual non-specific binding to the host DNA and/or chromatin-interacting proteins, though this has not been investigated so far.

To circumvent these issues, new enzymes were developed (reviewed in ref. [7]). BioID2 is derived from *Aquifex aeolicus* and is the smallest enzyme for PDB described to date (26.4 kDa)[4]. As a class I PBL, it also lacks a DNA-binding domain[7]. BASU is a variant of the *B. subtilis* BirA that lacks its DNA-binding domain and was claimed to show 1000-fold faster biotinylation kinetics than BioID[8], though this has been disputed[9]. A directed evolution approach led to TurboID and miniTurbo, two super active variants of BioID[9]. TurboID has the size of BioID and is the most active enzyme described to date supporting labeling kinetics down to 10 min[9]. This strong activity, however, comes at the cost of a high background labeling before the addition of extra biotin to the medium[9,10]. miniTurbo is smaller than TurboID and does not show high background activity. However, miniTurbo has about half the activity of TurboID[9] and was reported as unstable[10]. Furthermore, both TurboID and miniTurbo can lead to toxicity when constitutively expressed in mammalian cell culture or model organisms[9,10]. Finally, AirID is an enzyme faster than BioID and less toxic than TurboID but still requires hours of labeling time[11]. Altogether, each of these enzymes comes with advantageous properties and some drawbacks. Clearly, an enzyme that combines the most advantages would be a valuable tool for PDB.

Here we present two novel enzymes: microID, a truncation variant of BioID2, and its directed evolution-deduced variant ultraID. With a molecular weight below 20 kDa, microID and ultraID are by far the smallest PDB enzymes available to date. Moreover, ultraID exhibits enzyme kinetics similar to TurboID but with less background activity. UltraID supports efficient labeling in mammalian cell culture, *E. coli* and *S. cerevisiae*. In a direct comparison in which the protein Argonaute-2 (Ago2) was either fused to ultraID or BioID and expressed at physiological levels, we show that a 10 min labeling time with ultraID can substitute for overnight labeling with BioID. Finally, using ultraID, we probed the membrane-associated interactome of the COPI coat protein complex coatomer in living cells. Altogether, ultraID is the smallest enzyme for PDB and supports efficient short labeling.

## Results

### microID, a small size biotin ligase derived from BioID2. We previously developed a split-BioID assay in which BioID is split

into two inactive fragments that reassemble an active enzyme when fused to two interacting proteins[12]. We explored whether a similar assay could be set up with BioID2. Based on the structures of both proteins we split BioID2 between amino acids $K^{171}/S^{172}$ (Fig. 1a), corresponding to the split-BioID site ($E^{256}/G^{257}$). The resulting fragments NBioID2 (BioID2 [2–171]) and CBioID2 (BioID2 [172–233]) were then, respectively, fused to the proteins FKBP (12-kDa FK506-binding protein) and FRB (FKBP-rapamycin-binding domain) that interact in the presence of rapamycin[13]. To test split-BioID2, the fusion proteins were expressed in HeLa cells, and PDB was analyzed in the presence or absence of rapamycin (Fig. 1b). Biotinylation was observed irrespective of rapamycin treatment (Fig. 1c, lanes 1 and 2). In addition, CBioID2-FRB was hardly detectable and only in the presence of rapamycin (Fig. 1c, lanes 1 and 2), suggesting that it is unstable. This suggested that NBioID2, which contains the enzyme catalytic site, retains biotinylation activity. This is indeed the case as when NBioID2-FKBP was expressed in the absence of CBioID2-FRB, similar biotinylation activity was observed to when both NBioID2 and CBioID2 fusions were expressed (Fig. 1c, compare lane 3 to 1&2). NBioID2, with a size of 19.7 kDa, is the smallest enzyme for PDB to date and is termed from now on microID (or µID).

**microID shows efficient biotinylation at short labeling times**. We then analyzed how microID performs in comparison to other PDB enzymes. First, we compared BioID, BASU, and microID. After an overnight labeling (Supplementary Fig. 1, left), all three enzymes led to promiscuous biotinylation. By contrast only BASU and microID showed clear biotinylation after much shorter labeling times of 1 h (Supplementary Fig. 1, middle) and 10 min (Supplementary Fig. 1, right). Hence, in addition to its small size, microID is an enhanced activity enzyme in comparison to BioID.

In a second benchmarking round, we compared BioID2, BASU, TurboID, and microID (Fig. 2). We did not include miniTurbo as it was reported to be less active than TurboID[9] and highly unstable[10]. To assess the background activity of each enzyme, we compared the biotinylation obtained in the absence or presence of additional biotin for each timepoint. All four enzymes showed clear biotinylation over background levels at overnight (Fig. 2, left) and 1 h (Fig. 2, middle) labeling time. At 10 min labeling, the clearest biotinylation over background was observed for BASU, TurboID, and microID (Fig. 2, right).

Together, these experiments reveal microID as a novel small size enzyme that supports efficient PDB at short labeling times.

**Directed evolution of microID**. We next tried to improve the activity of microID through protein engineering. A single substitution, R118G, renders BirA of *E. coli* abortive. Situated in a conserved biotin/bio-AMP binding site, the corresponding mutation (R40G) had been introduced into BirA of *A. aeolicus* to obtain BioID2[4]. Substituting R118 with a serine instead of a glycine results in twofold increased activity in BioID[9]. We thus tested if the corresponding mutation, R40S, also leads to enhanced activity in microID. However, a side-by-side comparison revealed that R40S rather leads to weaker biotinylation when compared to R40G (Supplementary Fig. 2).

The hyperactive enzyme TurboID was obtained from a yeast display-directed evolution approach using BioID R118S as a starting point. We reasoned that we could engineer a higher activity enzyme using a similar strategy applied to microID. To this end, we generated, through error-prone PCR[14], a surface display library of approximately $8 \times 10^7$ random microID mutants with an estimated average mutagenesis rate of 2.2 amino acid exchanges per variant. The random variants were expressed

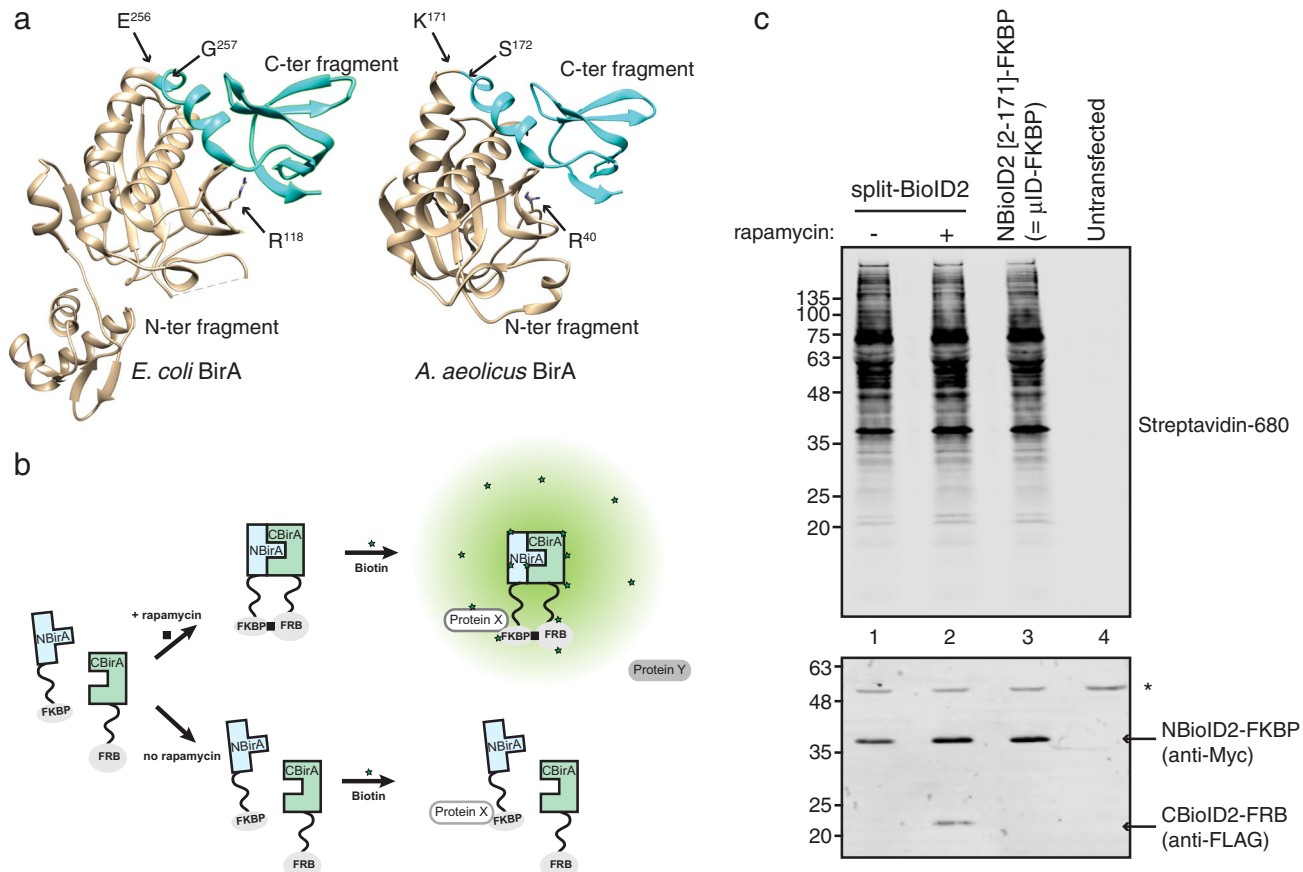

**Fig. 1 A truncation fragment of BioID2 with promiscuous biotinylation activity. a** *E. coli*'s and *A. aeolicus*' BirA crystal structures (Protein Data Bank 1HXD and 3EFS) showing the splitting site of split-BioID ($E^{256}/G^{257}$) and its corresponding position in BioID2 ($K^{171}/S^{172}$) as well as the mutated arginine ($R^{118}$ or $R^{40}$). **b** Principle of the rapamycin-induced dimerization set-up to test split-BioID fragments: FKBP is fused to the N-terminal fragment and FRB to the C-terminal fragment. PDB is tested in the presence or absence of rapamycin. **c** Blots of lysates of HeLa cells transiently transfected with the indicated constructs and cultivated overnight in biotin-containing medium in the presence or absence of rapamycin. Biotinylation was analyzed using IRDye680-labeled streptavidin and expression levels of the fusion proteins with antibodies against FLAG and Myc tag as indicated. The asterisk (*) indicates a non-specific cross-reactivity that was used as a loading control.

as 6xHis-tagged fusion proteins to Aga2p, a yeast protein that is exposed to the cell surface through its disulfide bonds linkage to Aga1p (Fig. 3a).

A cell surface biotinylation assay was performed by incubating the ligase-expressing cells with 50 μM biotin and 2.5 mM ATP for 17 h as a starting point. After cell staining using fluorescently labeled streptavidin (ligase activity) and anti-pentaHis antibodies (surface presentation), three consecutive FACS rounds with decreasing labeling times were conducted, sorting those cells that exhibited a strong fluorescence signal for biotin ligase activity and clear surface presentation (Fig. 3a, b). As a result, 0.77, 2.24, and 0.53% of the cells were enriched in rounds 1, 2, and 3, respectively. MicroID, which served as a control, showed negligible activity when the reaction time was shorter than 1 h (Supplementary Fig. 3). By contrast, after three rounds of enrichment, sorted cells showed significant activity after a 10 min labeling time (Fig. 3b). Eight individual clones of the final selection round were selected and analyzed by flow cytometry. The six clones that showed the strongest activity (Supplementary Fig. 4) were sequenced, revealing three individual mutants (Fig. 3c). Due to a single nucleotide deletion leading to a stop-codon loss, one microID variant was extended by 22 amino acids at its C-terminus (NSSRSDNNSVDVTKSTLFPLYF). Interestingly, the other two identified triple mutants shared an L41P mutation. This site is located next to the R40G mutation

within the biotin-binding site. All other substitutions are distributed at the surface of the enzyme, away from the catalytic site (Fig. 3d).

To investigate whether the observed enhanced cell biotinylation was the result of increased labeling of proximal proteins rather than enhanced self-biotinylation of the microID variants, enzymes were removed from the cell surface after the biotin ligase reaction by reducing the disulfide bonds between Aga1p and Aga2p with DTT. Surface exposition signals were considerably reduced after DTT treatment (Supplementary Fig. 5) while cell surface biotinylation persisted for both triple mutants but not for the C-terminally extended variant (Supplementary Fig. 5). Most likely, the C-terminal extension, which contains a lysine residue serves as an acceptor labeling substrate, eventually resulting in enhanced overall biotinylation. Therefore, this variant was excluded from further analysis. The triple mutants were named ultraID-4 (with the mutations S24C, L41P, K169R) and -5 (L41P, K115M, K156E).

The biotinylation activity of ultraID-4 and -5 was then assessed in HeLa cells after 10 min labeling time and compared to microID, the template for directed evolution, and TurboID, the most active enzyme described so far (Fig. 4a). Both variants yielded biotinylation signals stronger than that of microID and seemingly similar to that of TurboID. However, as enzyme expression differed between the ultraID clones and TurboID,

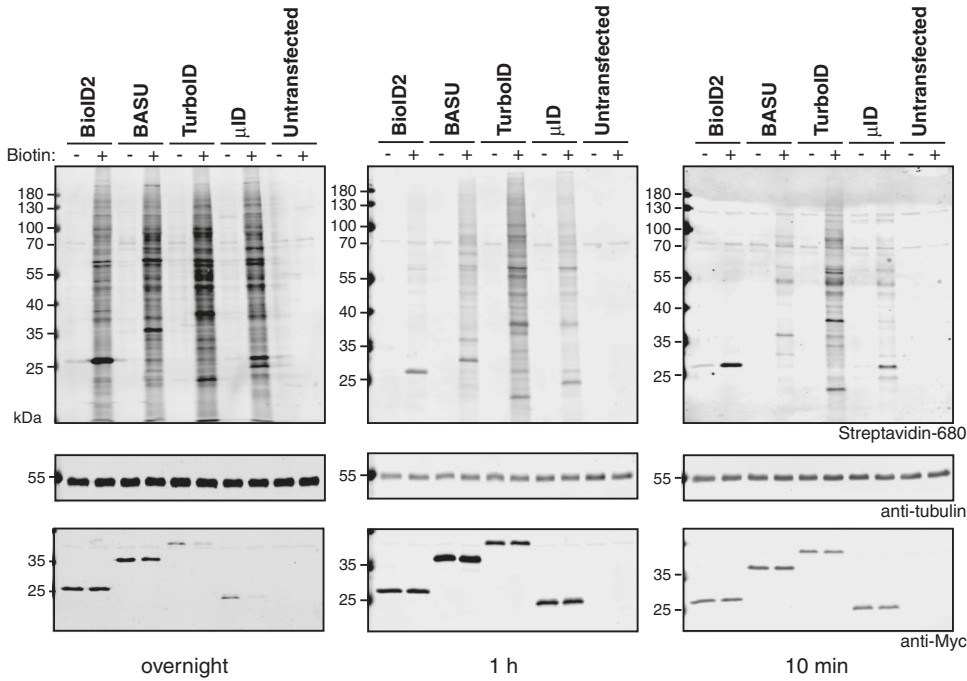

**Fig. 2 microID shows efficient biotinylation at short labeling times.** Blots of lysates of HeLa cells transiently transfected with the indicated constructs and incubated with 50 μM biotin overnight (left), for 1 h (middle), or for 10 min (right). Biotinylation was analyzed using IRDye680-labeled streptavidin and expression levels of the fusion proteins with antibodies against the Myc tag as indicated.

precise conclusions about their biotinylation activities were not drawn at this point.

**The mutation L41P is responsible for the enhanced activity of the ultraID variants**. The directed evolution strategy that led to TurboID resulted in 14 mutations on the original BioID sequence. None of them are close to the catalytic pocket and they do not cluster to a special part of the enzyme. Hence there is no obvious explanation on how these mutations lead to a higher activity apart from that they somehow indirectly affect the catalytical site[7,9]. The outcome of our directed evolution effort is quite different: ultraID-4 and -5 are triple mutants of microID that share an L41P mutation. Since L41 is adjacent to the R40G mutation within the biotin-binding site of microID, we reasoned that the enhanced activity of the ultraID variants can probably be mapped down to the single additional L41P mutation in microID. To address this, ultraID-4 and ultraID-5 as well as microID, microID R40G/L41P, and a microID L41P-only variant were produced as recombinant proteins in *E. coli* (Supplementary Fig. 6A). Enzymatic activity was then determined in an ELISA-based proximity-dependent biotinylation assay (see methods). In accordance to the single clone cell surface display analysis, ultraID-4 and -5 showed increased activity of 167 and 183%, respectively, compared to microID (Fig. 3e). Interestingly, the microID L41P-only mutant, with wild-type R40 position, showed a higher biotinylation activity than microID (Fig. 3e). Hence L41P, like R40G, is sufficient to render the BioID2 [2–170] fragment abortive. The effects of both mutations are additive as the microID variant with combined R40G/L41P mutations showed the highest activity of all tested variants (207%). Similar observations were made in HeLa cells in which both microID (R40G) and microID L41P-only (with intact R40 position) showed biotinylation activity after a 10 min incubation with biotin, and combining both mutations resulted in even stronger biotinylation signals (Fig. 4b).

To further characterize the enzymes, we determined their melting temperatures (Tm) with a thermal shift assay (see methods). Compared to microID (Tm = 64 °C), ultraID-5

showed decreased thermal stability (Tm = 52 °C) whereas ultra-ID-4 showed two independent melting points (55 °C and 64.5 °C), which could be a consequence of the S24C mutation that might promote the formation of an intermolecular disulfide bond. The microID L41P-only variant had a slightly decreased stability with a melting point of 60.5 °C whereas the double mutations R40G/L41P in microID-R40G/L41P did not impact its thermal stability (Tm = 63 °C) compared to microID (Supplementary Fig. 6b).

Altogether, the combination of R40G and L41P is the minimal set of mutations required for enhanced enzymatic activity, and the corresponding variant has a thermostability profile comparable to microID. The variant microID-R40G/L41P was thus selected for further characterization and is from here on referred to as ultraID.

**ultraID outperforms current enzymes for PDB**. Having defined ultraID with its minimal set of mutations, we compared its activity to that of other PDB enzymes. BioID, BioID2, BASU, TurboID, microID, and ultraID were expressed in HeLa cells, and the extent of protein biotinylation after overnight, 1 h, and 10 min labeling time was determined by western blot and compared to respective samples without biotin supplementation (Fig. 5a). Each experiment was replicated for quantitative estimation of the biotinylation activity. Three additional experiments for TurboID and ultraID with 10 min biotinylation were included in the quantitative analysis. The signals for streptavidin were integrated over each lane, excluding the band corresponding to the self-biotinylation of the ligases, and were normalized to the expression levels of each ligase (Myc signal).

All the ligases showed a clear activity over background biotinylation after overnight and 1 h labeling time with ultraID and TurboID having the strongest relative activity. With a 10 min labeling time, hardly any biotinylation over background levels was observed for BioID and BioID2 whereas BASU and microID still showed a clear activity. All the enzymes were, however, outperformed by TurboID and ultraID yielded 5–9 times higher biotinylation than microID and BASU under these conditions.

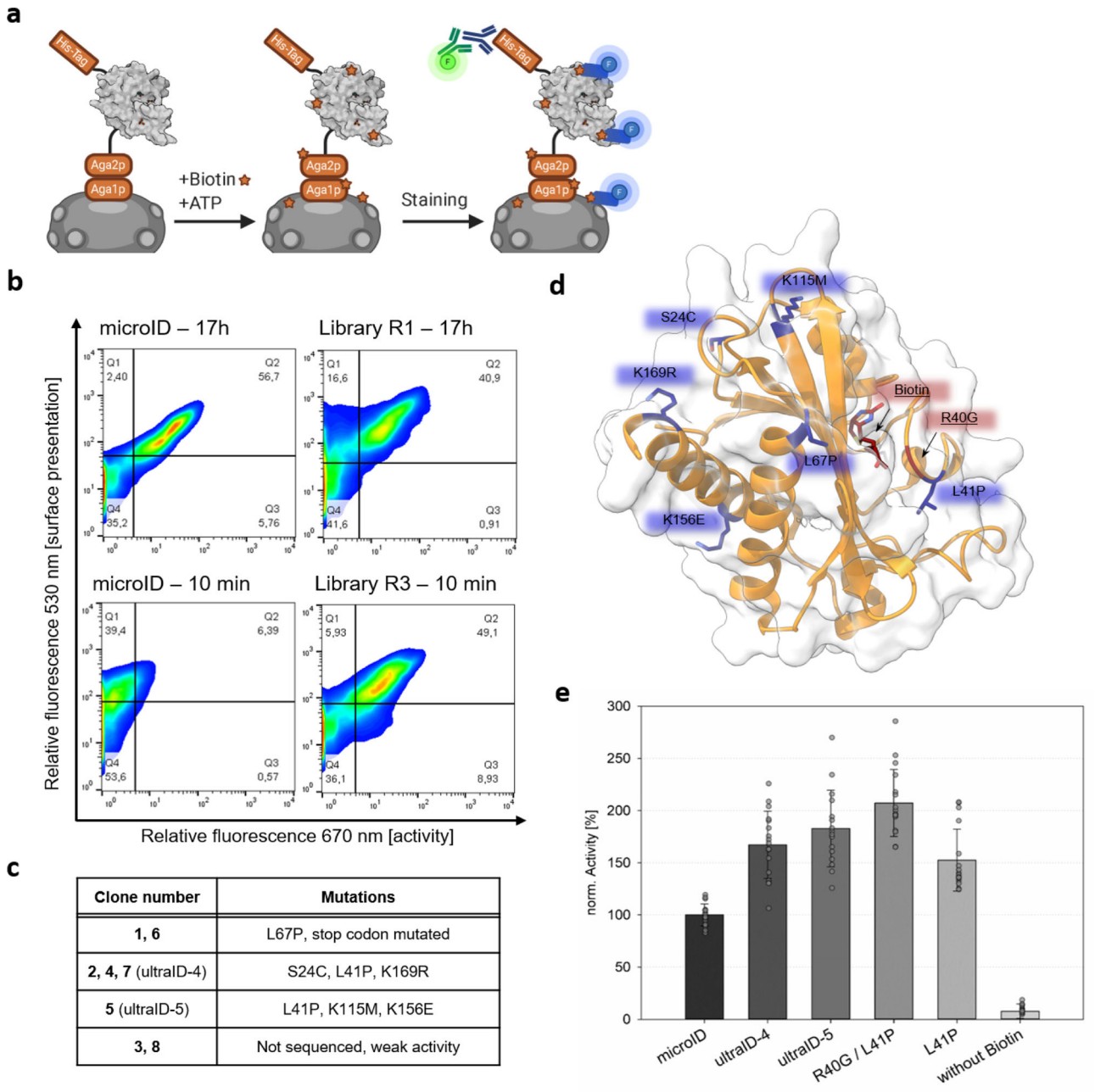

**Fig. 3 Directed evolution of microID. a** Scheme of the biotinylation assay. MicroID random variants were surface-presented via Aga1p:Aga2p. The yeast surface was biotinylated in presence of ATP and biotin. The His-tag served as a presentation marker. Biotin was detected with a streptavidin-APC conjugate. **b** Density plots of surface presentation (Y-axis) vs. cell surface biotinylation (X-axis). The error-prone PCR based-yeast surface display library was screened by stepwise decreased biotinylation time. Upper row: Comparison of microID with the initial library (R1) after a 17 h biotinylation assay. Bottom row: same with the Round 3 (R3) library after a 10 min biotinylation assay. **c** Single clone analysis from the Round 3 library. **d** Positions of the mutated residues (structure based on PDB: 3EFR processed with the program ChimeraX). **e** Activity measurement of microID variants with an ELISA-based biotinylation assay. Measurements were performed in five individual experiments in triplicates or quadruplicates. Norm. activity refers to the microID absorbance at 405 nm set to 100%. Error bars indicate standard deviations.

The average relative signal for TurboID in the biotin-treated sample after a 10 min pulse labeling was on average ca. 25% higher than that for ultraID, though this was not consistent in all experiments (Fig. 5b). In accordance with the previous reports[9,10], the high activity of TurboID came, however, at the cost of a high background signal from the sample without biotin addition that was consistently observed in all experiments and on average almost 3 times higher than for ultraID (Fig. 5a, b).

Together, our data show that ultraID yields a labeling efficiency similar to TurboID at labeling times down to 10 min, with the

potential advantage of a lower background biotinylation activity, making it a more efficient enzyme for PDB.

**PDB with microID and ultraID in yeast and bacteria**. An important feature of novel enzymes for PDB is their suitability for various model organisms. As the directed evolution of microID was based on yeast display, it was expected that the enzymes are also active in *Saccharomyces cerevisiae*. As we previously applied BioID in yeast to probe the microenvironment of the ribosomal

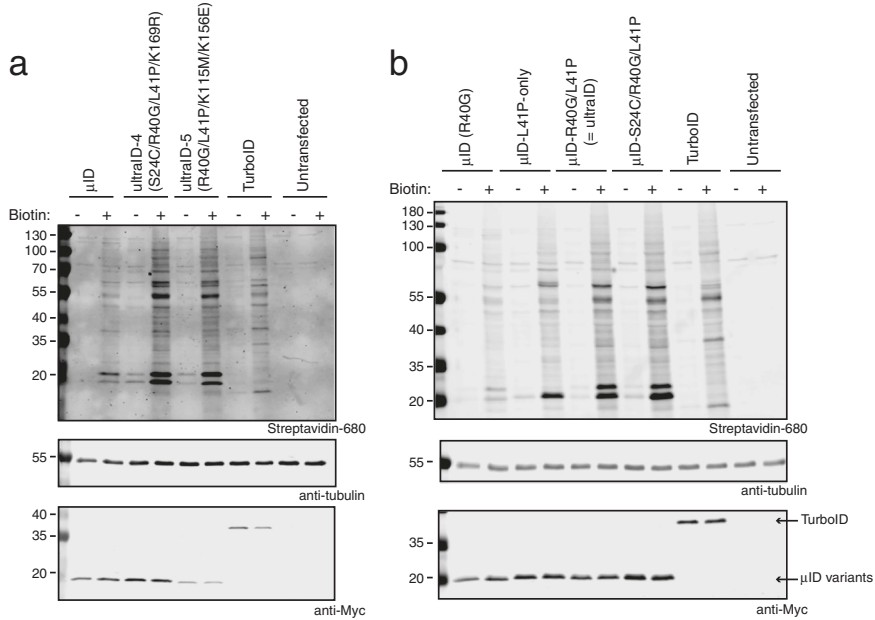

**Fig. 4 ultraID variants show improved activity compared to microID.** Blots of lysates of HeLa cells transiently transfected with the indicated constructs and incubated with 50 µM biotin for 10 min. Biotinylation was analyzed using IRDye680-labeled streptavidin and expression levels of the fusion proteins with antibodies against the Myc tag as indicated. **a** Comparison microID, ultraID-4 and -5, and TurboID. **b** Comparison microID, microID L41P-only, microID R40G/L41P (=ultraID), microID S24C/R40G/L41P (=ultraID-4 with K169R reverted to WT) and TurboID.

protein Asc1/RACK1[15], experiments in *S. cerevisiae* were conducted in which BioID, TurboID, microID, and ultraID were fused to Asc1. Comparisons of the four enzymes showed that while they all work in yeast, TurboID, microID, and ultraID showed stronger activity than BioID (Fig. 6a). We note, however, that in *S. cerevisiae* expression of the microID, ultraID, and especially TurboID fusion proteins all lead to background activity from the low biotin concentrations (8.2 nM) in the YNB medium without additional biotin (- condition on Fig. 6a).

We also tested the enzymes in *E. coli*. Also in this organism, a clear biotin-dependent labeling activity was observed upon expression of TurboID, microID, and ultraID (Fig. 6b).

In conclusion, both microID and ultraID are well suited for PDB in bacteria, yeast, and mammalian cell culture.

**UltraID supports PDB-MS with a 10 min labeling time.** To investigate the performance of ultraID in authentic conditions, in which the enzyme is fused to a target protein expressed at physiological level, we then established doxycycline-inducible HeLa cell lines that express ultraID fused to the N-terminus of Argonaute-2 (Ago2) from a single gene locus as we described previously[12,16]. To filter background biotinylation from the proteomic data, we constructed an additional control cell line that expressed ultraID fused to the unrelated protein Rab11. Rab11 was chosen as a negative control as, similar to Ago2, it is in equilibrium between the cytosol and endomembranes[17].

As a comparison, stable cell lines for TurboID-Ago2 and TurboID-Rab11 were also constructed, and cell lines for BioID-Ago2 and BioID-Rab11 were already available in our laboratory[12]. Expression levels were tuned so that the Ago2 fusion proteins were expressed close to endogenous levels (Supplementary Fig. 7) and Ago2 and Rab11 fusions yielded similar biotinylation levels across cell lines. The fusion proteins showed a localization pattern typical for Ago2 and Rab11 in indirect immunofluorescence experiments (Supplementary Figs. 8 and 9). PDB was then performed upon addition of exogenous biotin for 10 min (ultraID and TurboID fusions) or overnight (BioID fusions) to four biological replicate samples. The resulting

biotinylated proteins were processed for MS analysis after their capture with trypsin-resistant streptavidin beads[18] and on-beads trypsin digestion (Fig. 7a). Peptide identification was performed with the MaxQuant software[19] and the relative enrichment of each protein across samples was quantified by label-free quantification (LFQ)[20] using the proDA (probabilistic drop-out analysis) package to infer for each protein its mean LFQ intensity and associated variance, taking into account the missing values[21]. To obtain the proxiome of Ago2, we tested the differential abundance of proteins between the Ago2 and Rab11 datasets by comparing the inferred LFQ mean intensities for the two conditions. When examining the data, we noticed that the LFQ-based enrichment analysis occasionally missed some valid candidates that were detected in all Ago2 replicate samples but in none of the Rab11 control samples. We thus also added to the list of true-positive hits those proteins that did not pass the thresholds of the LFQ analysis but fulfilled the following criteria: detection in all four replicates of Ago2 samples with at least two unique peptides in three replicates and no peptide detected in any Rab11 sample. The complete list of hits for the BioID, ultraID, and TurboID datasets is provided in Supplementary Data 1 and 2.

The ultraID-derived Ago2 interactome obtained after 10 min labeling was somewhat smaller (50 vs. 68 hits) than that obtained from BioID after overnight labeling (Fig. 7b) but ultraID performed equally well to BioID in identifying known Ago2-associated proteins such as TNRC6 proteins[22], Dicer[23], CNOT1[24,25], GIGYF proteins[12] or XRN1[26] (Fig. 7b) and the volcano plots resulting from both datasets were similar (Supplementary Fig. 10). We obtained a smaller number of hits (17) from the TurboID dataset but they also represented relevant proteins (Fig. 7b). This small number was not expected but may be due to the use of another batch of streptavidin beads for this series of pulldowns. The subset of preys specific to the BioID, ultraID or TurboID datasets also contains relevant proteins, such as CNOT11, DCP1A/B, PABPC1/4 for BioID; FXR1/2, PUM1 for ultraID, and CNOT8 for TurboID (Supplementary Data 1 and 2). Hence a labeling bias towards a certain type of protein and/or localization is not obvious from our data.

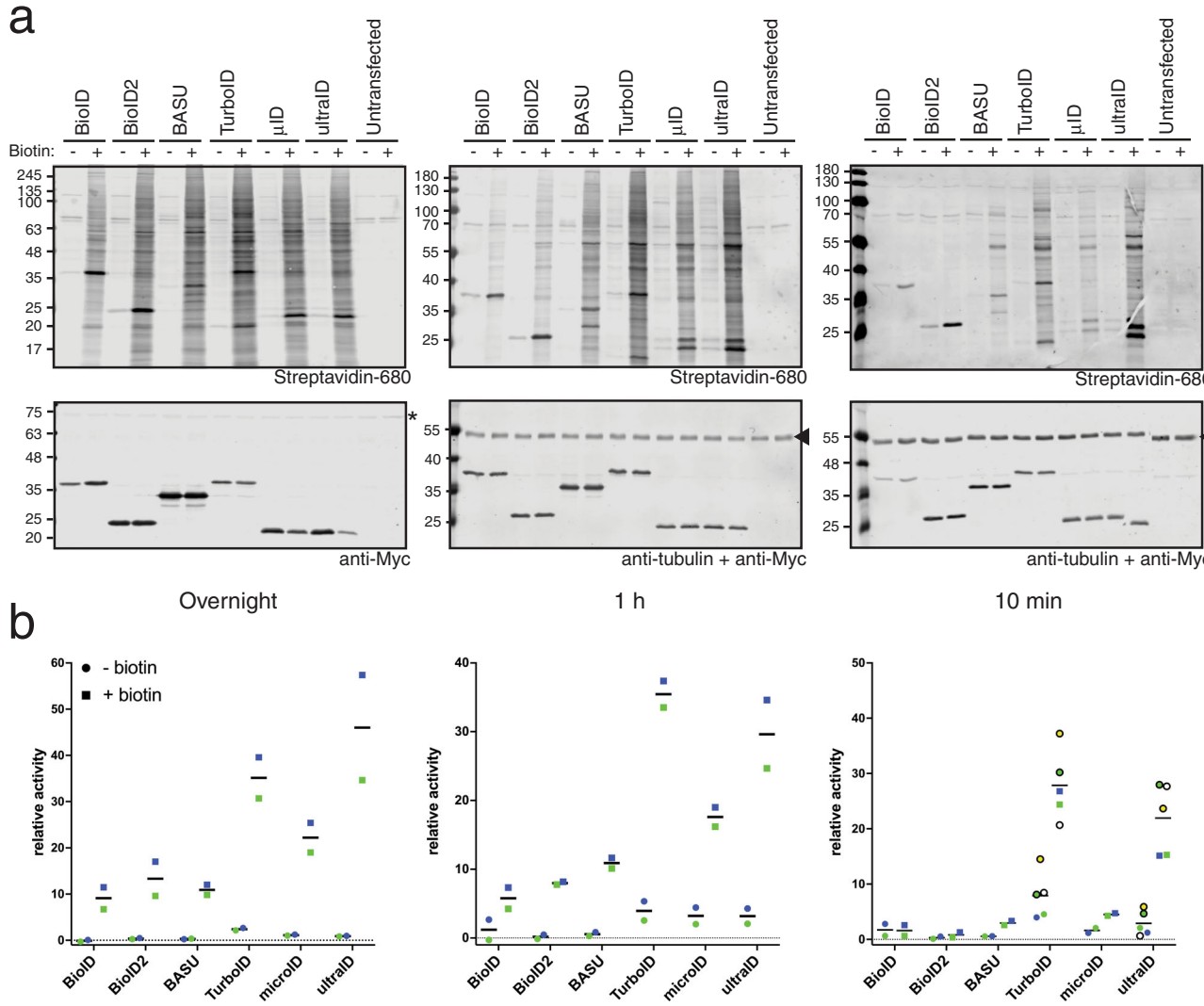

**Fig. 5 ultraID outperforms current PDB enzymes. a** Blots of lysates of HeLa cells transiently transfected with the indicated constructs and incubated (+) or not (−) with 50 µM biotin overnight (left), for 1 h (middle), or for 10 min (right). Biotinylation was analyzed using IRDye680-labeled streptavidin and expression levels of the fusion proteins with antibodies against the Myc tag as indicated. The asterisk (*, overnight condition) indicates a non-specific cross-reactivity that was used as a loading control. The arrowhead (1 h and 10 min conditions) indicates the tubulin signal that was used as a loading control. **b** Quantification of **a**. Circles and squares of the same color belong to the same replicate experiments.

To compare the background labeling activity of ultraID and TurboID prior to biotin addition, we also performed streptavidin pulldowns with the same batch of beads from cells that were not incubated with extra biotin. In the corresponding MS datasets, the number of hits with consistent detection in all three TurboID/ultraID-Ago2 replicates was small (respectively 14 and 13 hits with at least two assigned LFQ values across replicates) and the proDA analysis did not converge. To assess the labeling background, we thus analyzed the abundance of the fusion proteins and two core interacting partners in each sample using the iBAQ (intensity Based Absolute Quantification) algorithm[27]. The abundance of Ago2 and its two interacting partners TNRC6A and TNRC6B[28] was lower (at least 15 fold) in the pulldown from the ultraID-Ago2 cell line than from the TurboID-Ago2 cell line. Consistently the abundance of Rab11 was 15-fold lower in the pulldown from the ultraID-Rab11 cell line than from the TurboID-Rab11 cell line and the two core interacting partners of Rab11, RAB11FIP1 and RAB11FIP5[17], were not detected at all in the ultraID-Rab11 cell line (Fig. 7c and Supplementary Data 2).

Together, our data show that ultraID allows obtaining relevant proteomic datasets in PDB experiments with a labeling time of 10 min at physiological expression levels and may provide a favorable balance of signal vs. background when compared to TurboID.

**Defining the membrane-associated interactome of coatomer with ultraID.** To further validate the practicality of ultraID we analyzed a membrane-associated interactome of coatomer, the COPI coat protein complex. During the formation of COPI transport vesicles, the small GTPase Arf1 (or one of its paralogs) recruits coatomer to the Golgi membrane when bound to GTP. There, coatomer polymerizes and together with Arf1 promotes membrane deformation, cargo protein capture, and membrane scission. As a cargo-loaded transport vesicle sheds from its donor membrane, GTP-hydrolysis by Arf1, stimulated by its ArfGAPs, leads to the release of coatomer to the cytosol[29] (Supplementary Fig. 11). Because the active form of coatomer is the population that transiently associates with endomembranes, it is not

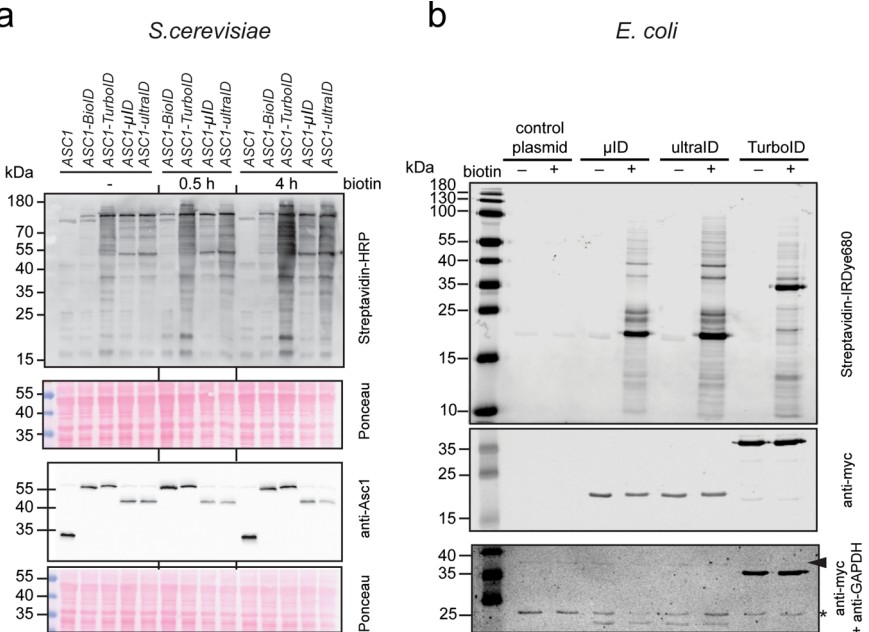

**Fig. 6 PDB in yeast and bacteria with microID and ultraID. a** Blots of lysates of *S. cerevisiae* strains expressing the indicated Asc1 fusion proteins and incubated or not (−) with 10 μM biotin for the indicated times. The fusion proteins were expressed from plasmids in a Δ*asc1* strain, and the *ASC1* wild-type strain carrying the empty vector served as a control. Biotinylation was analyzed using HRP-labeled streptavidin and expression levels of the fusion proteins with antibodies against Asc1 as indicated. **b** Blots of lysates of *E. coli* cells transformed with expression plasmids for ultraID, microID, TurboID, or the control plasmid. The cells had been incubated (+) or not (−) with biotin for 16 h. The arrowhead indicated the expected size for bacterial GAPDH, and the asterisk (*) indicates a non-specific cross-reactivity that was used as a loading control.

amenable to proteomic studies using affinity purification approaches. Defining a relevant interactome of coatomer has thus been challenging. Currently available datasets came from cellular fractionation assays or reconstitution experiments in which COPI vesicles were purified, out of their native context under non-physiological conditions[30,31]. To probe the interactome of coatomer in living cells, we used ultraID as a C-terminal fusion to the γ-COP subunit of coatomer. Of the seven coatomer subunits, we picked γ-COP because it can be tagged at its C-terminus without affecting its functionality[32] and because this subunit exists as two paralogs, γ1-COP and γ2-COP, with overlapping but not fully identical functions[32,33].

To ensure a physiological relevance of the datasets, γ1-COP-ultraID and γ2-COP-ultraID were expressed as inducible rescue constructs respectively in γ1-COP and γ2-COP knock-out P19 mouse pluripotent cells[32], with their expression levels tuned to those of the endogenous proteins (Fig. 8a). To filter background biotinylation from the proteomic data, we constructed an additional control cell line that expressed ultraID fused to Ago2 (Fig. 8a). Ago2 was chosen as a negative control as it is not functionally related to the COPI pathway but has a reported Golgi/ER partial localization[34]. The drug brefeldin A (BFA) inhibits the activation of Arf1 at the Golgi membrane and thus prevents the association of coatomer with membranes[35,36] (Supplementary Fig. 11), which leads to the collapse of the Golgi into the ER[37]. In indirect immunofluorescence experiments γ1-COP-ultraID and γ2-COP-ultraID showed the expected BFA-sensitive Golgi localization assessed by co-localization with the Golgi marker protein GM130 (Supplementary Fig. 12 upper and middle panels). By contrast, in both mock- and BFA-treated cells, Ago2-ultraID showed a punctuated staining pattern (Supplementary Fig. 12 lower panel) reflecting the partial localization of Ago2 to P-bodies[38]. Moreover, whereas in the mock condition biotinylated proteins by γ1-COP-ultraID and γ2-COP-ultraID after PDB were concentrated in the Golgi area, they showed a

diffuse cytosolic staining under BFA-treatment (Supplementary Fig. 12).

PDB was then performed on the three cell lines under mock- or BFA-treatment conditions (for the γ1-COP-ultraID and γ2-COP-ultraID cell lines) with three biological replicates for each condition (total of 15 samples, see Fig. 8b). To assess the specificity of the proteomic datasets obtained after a longer labeling time with ultraID, cells were incubated with biotin for 4 h. The proteomic data were analyzed as described above. We first considered the γ1-COP and γ2-COP proxiomes under mock-treated conditions in which coatomer is present in both its membrane-associated and cytosolic forms (Supplementary Fig. 13 and Supplementary Data 3) and identified 56 and 53 proximal proteins to γ1- and γ2-COP respectively. Of these, 59% of γ1-COP proximal proteins and 42% of γ2-COP's have a known Golgi localization. Among them, we found as expected the other COP subunits of the coatomer complex, the paralogs of the small GTPase Arf1 and their three ArfGAPs.

To gain a better insight into the potential interactome of membrane-associated coatomer we considered the proteins enriched in the mock condition compared to the BFA-treated samples in which coatomer is released from Golgi membranes (Fig. 8c). Proteins that are enriched in the mock samples are indeed likely to represent membrane-associated proteins proximal to coatomer. With this analysis, we identified 21 and 17 proximal proteins to membrane-associated γ1- and γ2-COP respectively (Fig. 8d and Supplementary Data 3). Remarkably, all but one are known Golgi proteins demonstrating the specificity of the assay. The only non-Golgi protein is emerin, a membrane-anchored protein of the inner nuclear membrane (INM). Another INM membrane protein, Sun2, uses an arginine-based COPI-binding signal that mediates retrieval from the Golgi and participates in nuclear localization. Such an arginine motif is also found on other INM proteins including emerin[39]. As a partial Golgi localization has been described for emerin[40], there

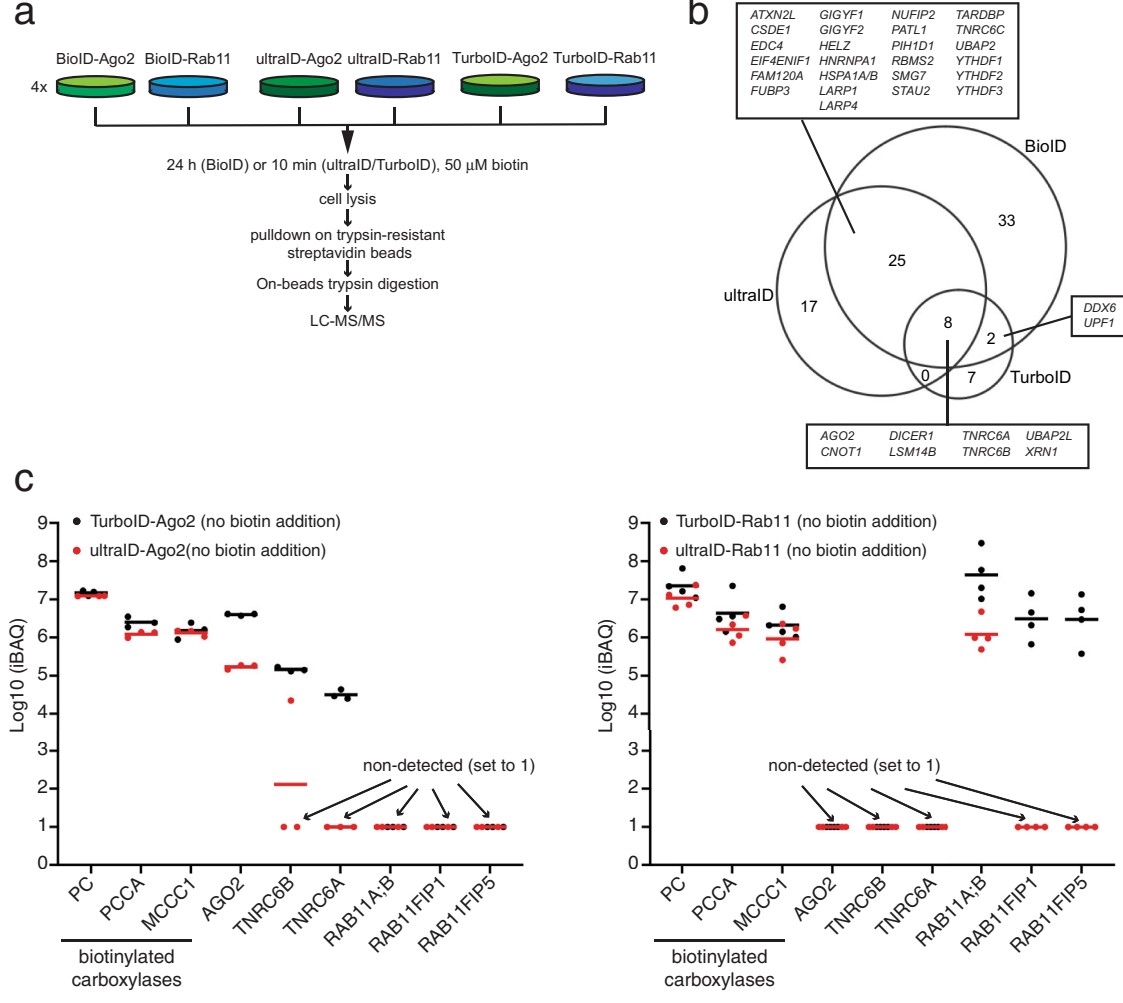

**Fig. 7 Ten-min labeling with ultraID under experimental PDB conditions. a** Experimental set-up for the side-by-side comparison of overnight labeling with BioID vs. 10 min labeling with ultraID or TurboID. **b** Venn diagram displaying the proteomic dataset sizes and overlap of the significant hits from **a**, the gene names of the dataset overlap are specified. **c** Relative abundance, assessed with iBAQ values, of endogenous biotinylated carboxylases and main interacting partners of AGO2 (TNRC6A & B) and RAB11A (RAB11FIP1 & 5), obtained from HeLa cells expressing TurboID-Ago2 and ultraID-Ago2 (left) or TurboID-Rab11and ultraID-Rab11 (right) after streptavidin pulldowns with no biotin addition. Bars are mean values, $n = 3$ (Ago2 cell lines) or 4 (Rab11 cell lines).

may be a substantial amount of that protein at the Golgi in P19 cells that, like Sun2, relies on COPI vesicles to reach the INM.

**ultraID identifies expected membrane-associated interactors of coatomer.** Unsurprisingly, Arf proteins and their ArfGAPs, which are respectively responsible for coatomer recruitment to and release from membranes[29], were identified as membrane-associated interactors of γ1- and γ2-COP. Golgins are membrane-anchored coiled-coil proteins that tether specific transport vesicles to distinct regions of the Golgi apparatus[41]. In cooperation with additional proteins, golgins then stimulate the fusion of the tethered vesicles with their target membrane by allowing the formation of a tight SNARE complex[42]. Accordingly, the SNAREs syntaxin-5 and GOS-28, both involved in COPI-dependent intra-Golgi trafficking[43], and SLY-homolog, a protein that is required for Golgi SNARE pairing[44], were identified in the membrane-associated datasets. The same goes for the COPI-interacting golgin tethers giantin[45], TMF[46], and golgin-84 (ref. [47]). We also identified trans-Golgi network (TGN) localized golgins (GCC88, GCC185, and golgin-245) as membrane-associated interactors of coatomer. This was not expected as these are normally involved in tethering vesicles coming from endosomes[48]. Combined with the recent identification of a COPI-

mediated transport route from endosomes to TGN in yeast[49], this calls for further investigating a role of the COPI pathway in endosomal trafficking in mammals that was suggested by several reports more than 20 years ago[50,51].

ERGIC-53 is a cargo receptor that constitutively cycles, through COPII and COPI vesicles, between the ER and the ER-Golgi intermediate compartment (ERGIC), and can be found at low concentrations at the *cis*-cisterna of the Golgi but not beyond[52]. Transport vesicles that act at the *cis*-side of the Golgi are tethered by golgin-84[48]. Interestingly, we identified both ERGIC-53 and golgin-84 as specific membrane-associated proximal proteins to γ1-COP. This is in agreement with the observation that γ1-COP-containing coatomer is preferentially localized to the *cis*-side of the Golgi, whereas γ2-COP-containing coatomer localizes preferentially at the *trans*-side[33].

In summary, ultraID labeling allowed us to define a membrane-associated interactome of coatomer that faithfully reflects its expected proximal landscape.

**Discussion**

Here we describe ultraID, a novel engineered enzyme for efficient PDB in living cells. UltraID allows robust biotinylation within short labeling times to levels similar to TurboID. To date, ultraID

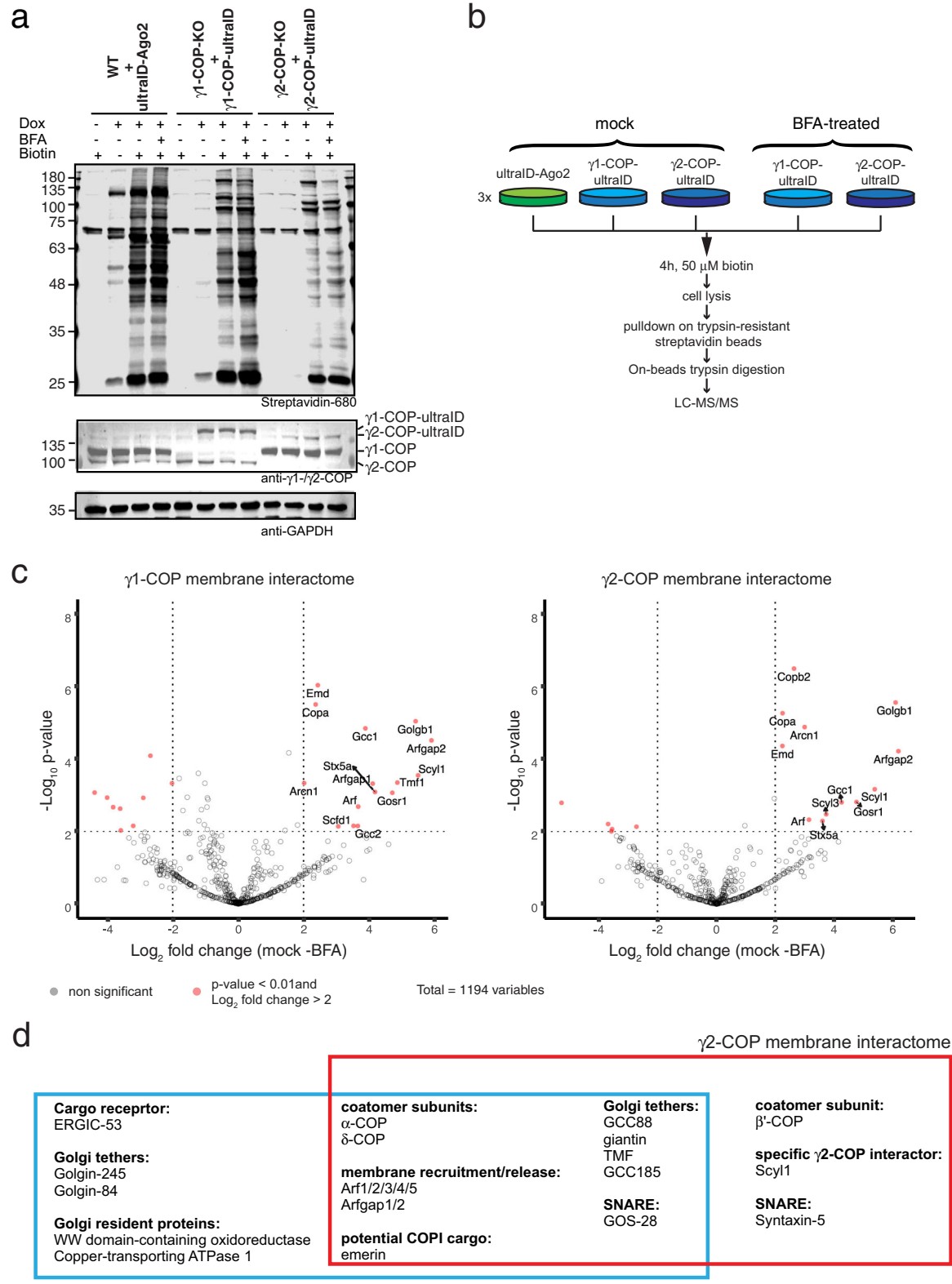

**Fig. 8 Probing a membrane-associated interactome with ultraID. a** Blots of lysates of P19 cells stably expressing ultraID-Ago2, of P19 cells knock-out for endogenous γ1-COP and rescued by γ1-COP-ultraID, and of P19 cells knock-out for γ2-COP and rescued by γ2-COP-ultraID lines. The fusion proteins were detected with antibodies against γ1- and γ2-COP and biotinylation with IRDye680-labeled streptavidin. **b** Experimental set-up for the determination of the membrane-associated interactome of γ1-COP and γ2-COP. **c** Volcano plots of the proteins identified by LC-MS/MS with the mock vs. BFA samples. Significant hits (*p* value <0.01 and log$_2$ fold change >2 in this analysis, and *p* value <0.01 and log$_2$ fold change >2 in the γ-COP vs. Ago2 comparison under mock conditions), are indicated with gene names in red. **d** Proteins identified as membrane interactors of γ1-COP and γ2-COP (see text).

is by far the smallest enzyme for PDB, 44% smaller than TurboID/BioID and 25% smaller than BioID2 (19.7 vs. 35 and 26.4 kDa, respectively). This may also be advantageous as it is generally considered that the larger the size of a tag, the greater the chance of interfering with the function, trafficking, and interactions of the fusion protein[53]. For example, BioID, which has the same size as TurboID, prevents the correct localization of the protein Sun2 to the INM[4], a process that is sensitive to the size of the fusion partner[54]. By contrast, BioID2, which is larger than ultraID, leads to a correct INM localization when fused to Sun2[4]. UltraID also produces less labeling background than TurboID when no exogenous biotin is added. This may prove important in some applications as the background activity of TurboID can lead to a considerable loss of specificity[10].

Using ultraID, we analyzed the remodeling of the interactome of COPI coat proteins after incubation with BFA. This allowed us to define a membrane-associated interactome. As transport vesicles are transient carriers that rely on dynamic protein/membrane interactions for their biogenesis and functions, defining their proteome is a challenging task. Previously, a landmark core proteome of COPI vesicles was defined based on the LC-MS analysis of isolated COPI vesicles generated in vitro from permeabilized cells and recombinant coat proteins[31]. A striking difference between this dataset and ours is the presence of glycosylation enzymes and cargo receptor proteins in the former, and the presence of many golgins and the membrane recruitment machinery in the latter. This illustrates the advantages and drawbacks of each approach. The in vitro isolation removes vesicles from their native cellular environment. As a consequence, it is less likely to capture the interactions that occur at the Golgi membrane. The PDB approach allowed us to probe the direct neighborhood of γ1- and γ2-COP in living cells, but it also limited our ability to analyze the content of COPI vesicles. Indeed, soluble cargo proteins inside the vesicles are inaccessible to bioAMP released in the cytosol. Similarly, transmembrane COPI cargo proteins such as Golgi-resident glycosylation enzymes typically expose short tails to the cytosol whereas the largest parts of the proteins are inserted in the lumen where they exert their functions[55]. This limits the amount of potentially available acceptor lysine residues and thus the likelihood for biotinylation to occur. All in all, this stresses that complementary approaches are needed to comprehensively define the proteome and interactome of transport vesicles.

Potential specific functions of the γ1- and γ2-COP paralogs have remained elusive. In vitro data suggest functional redundancy[31,56] but cellular assays point to some differences[32,33]. Notably, we showed that γ1-COP plays a unique role in promoting neurite outgrowth during the neuronal differentiation of P19 cells[32]. The molecular mechanisms underlying this observation are currently unknown. Having established ultraID-tagged P19 cell lines, it will now be possible to analyze the interactome of γ1- and γ2-COP during neuronal differentiation and possibly to identify differences that may help understand their specific functions in this context.

To summarize, we have developed ultraID a compact and hyperactive enzyme for PDB that favorably compares to previously available BPLs. Ultimately, ultraID is the product of two engineering steps: the removal of the C-terminal domain of the class I BPL BioID2, and the addition of one point mutation (L41P) to the canonical R40G substitution within the biotin-binding domain. As removal of the C-terminal domain of BioID yields a fragment with hardly any detectable activity[12], it was not expected that microID would be even more efficient than BioID2. A possible explanation may be the origin of BioID2. Coming from an extremophile organism, its overall structure is likely more rigid than that of BioID, possibly making its core catalytic

domain less prone to destabilization when the C-terminal domain is removed. The position of the L41P substitution suggests that it strengthens the destabilization of the biotin-binding site induced by the R40G substitution, thereby stimulating the release of bioAMP. This is supported by the observation that L41P alone is sufficient to render the truncated BioID2 [2–171] abortive and that the effects of R40G and L41P are additive. The engineering of ultraID may thus provide a simple framework to improve the activity of PDB enzymes based on class I BPLs. It also suggests that specifically targeting the biotin-binding site of BPLs for random mutagenesis may be a promising approach to generate hyperactive enzymes.

## Methods

**Plasmids and antibody**. The plasmids, yeast strains, primers, and antibodies used in this study are provided in Supplementary Tables 1–4. Plasmids for bacterial and mammalian expression of microID and ultraID are available at Addgene (Plasmid IDs 172878–172881).

**Mammalian cell culture**. All cell culture experiments were conducted with low-passage HeLa 11ht cells[57] obtained from Dr. Kai Schönig (Zentralinstitut für Seelische Gesundheit Mannheim, Germany) or P19 cells obtained from Sigma-Aldrich. HeLa cells were maintained as described in ref. [12] and P19 cells as described in ref. [32]. The cells were regularly tested for mycoplasma contamination.

**Construction of stable cell lines**. Flp-recombinase-mediated construction of stable HeLa 11ht cell lines was performed as described in ref. [57]. P19 stable cell lines were constructed using the Piggy-BAC transposon system as described in ref. [32].

**Immunofluorescence microscopy**. 9'000 HeLa cells or 20'000 P19 cells per well were seeded in 8-well μ-slides (Ibidi). On the following day, the expression of the ligase fusion proteins was induced with doxycycline. For BioID HeLa cell lines, 50 μM biotin was added at the same time as doxycycline, and cells were incubated at 37 °C for 24 h. For ultraID HeLa cell lines, the biotinylation time was 10 min one day after induction. For P19 cells, BFA-treated samples were first incubated for 1 h with BFA and then for an additional 4 h with BFA and biotin. The mock samples were only incubated with biotin for 4 h. After biotinylation, cells were carefully rinsed with ice-cold PBS, subsequently fixed and permeabilized with 150 μL ice-cold methanol, and incubated at −20 °C for 10 min. All further steps were performed at RT. Methanol was removed and the cells were washed three times with PBS. The samples were blocked with PBS + 5% bovine serum albumin (BSA, Carl Roth) (blocking solution) for 15 min. Incubation with the relevant primary antibodies was performed for 30 min at RT or overnight at 4 °C. The cells were washed three times with a blocking solution and incubated with the secondary antibodies and streptavidin-AlexaFluor647 for 30 min at RT. Then, the cells were washed three times with PBS. After a 10 min staining with DAPI (0.1 μg/mL in PBS) and brief washing with water, the cells were mounted with Mowiol mounting medium. After incubation at RT overnight, the slides were stored at 4 °C until imaging. Microscopy images were acquired with a Nikon Eclipse Ti2 spinning disk confocal microscope (Nikon Imaging Center, Heidelberg) using the acquisition software Volocity 6.3 (Perkin Elmer).

**Screening for biotinylation**

*In HeLa cells*. Screening for biotinylation was performed as described in refs. [58,59].

*In S. cerevisiae*. Cells were transformed with the pME4478 (*ASC1-birA\**), pME5479 (*ASC1-TurboID*), pME5086 (*ASC1-μID*) and pME5087 (*ASC1-μltraID*) plasmids and cultivated in yeast nitrogen base medium (YNB: 1.5 g/L YNB without amino acids, 5 g/L ammonium sulfate, 2% glucose) with 20 mg/L L-tryptophan at 30 °C. Yeast strains used in this work are listed in Supplementary Table 2. After incubation with biotin (50 μM), cells were lysed through alkaline treatment and subsequent boiling in SDS-PAGE loading buffer[60]. An equal number of cells of each sample (volume of culture harvested in mL = 5 divided by OD$_{600}$ of the culture) was used to generate cell lysates. The cells were harvested by centrifugation, washed with water, and resuspended in 400 μL 0.1 M NaOH. After incubation for 5 min at room temperature, samples were centrifuged, supernatants removed, and 100 μL of 1:4 diluted loading dye (0.25 M Tris-HCl pH 6.8, 30% glycerol, 15% β-mercaptoethanol, 7% SDS, 0.3% bromophenol blue) added. Samples were incubated for 3 min at 95 °C, centrifuged at 16,200 × *g* for 5 min, and 5 μL of the supernatants were subjected to SDS-PAGE followed by protein transfer onto nitrocellulose membrane. Proteins on the membrane were stained with Ponceau Red (0.2% PonceauS, 3% trichloroacetic acid), and blocked with either 5% milk powder or 1% BSA dissolved in phosphate-buffered saline (PBS). The membranes were then incubated with polyclonal rabbit anti-Asc1 (in 5% milk powder, 0.1% Tween 20, PBS) followed by incubation with a peroxidase-coupled secondary antibody or with

Pierce™ High Sensitivity Streptavidin-HRP (Thermo Fisher, #21130, diluted 1:2000 in 1% BSA, 0.1% Tween 20, PBS). Chemiluminescent signals were detected with the FUSION-SL-4 imaging system (Peqlab).

*In E. coli.* For each sample, one colony of T7 Express competent cells (NEB), transformed with a pET-15b plasmid for the expression of microID, ultraID, or CNOT9, was used to inoculate 5 mL LB medium containing 100 µg/mL ampicillin (LB-amp). After overnight incubation at 37 °C, the samples were diluted to an $OD_{600}$ of 0.6 in 6 mL LB-amp medium. Protein expression was then induced with 1 mM isopropyl β-D-1-thiogalactopyranoside (IPTG) for 1 h and then the cultures were split into two tubes with 3 mL culture volume. For each strain, biotin was added to one of those two tubes at a concentration of 50 µM while the other tube without biotin served as a negative control. The cultures were then incubated at 37 °C for another 16 h. Thereafter, $OD_{600}$ was determined and volumes with cell numbers equivalent to 200 µL of $OD_{600} = 0.6$ were centrifuged at $4000 \times g$ for 10 min. The cell pellets were resuspended in 50 µL 5× sodium dodecyl sulfate (SDS) loading buffer (5% β-mercaptoethanol, 0.02% (w/v) bromophenol blue, 10% SDS, 30% glycerol, 250 mM Tris-HCl pH 6.8), incubated at 95 °C for 5 min and subsequently diluted to 1× with 200 µL water. 10 µL of these dilutions were used for western blot analysis.

**Western blot analysis.** For each sample, 10 µg of protein extract in 1× SDS loading buffer was loaded onto a discontinuous SDS polyacrylamide gel. After electrophoresis, proteins were transferred to a nitrocellulose membrane (Cytiva) by wet blotting at 4 °C for 1 h. Subsequent steps and fluorescence-based western blot quantitative analysis were performed as described in ref. [12] with the LI-COR Image Studio software. Full scans of the blots used for the figures are provided in Supplementary Figs. 14–16.

**Preparative proximity-dependent biotinylation.** PDB in HeLa cell lines was performed with four biological replicates for each sample (ultraID-Ago2, ultraID-Rab11, TurboID-Ago2, TurboID-Rab11, BioID-Ago2, and BioID-Rab11). Cells were seeded in 15 cm dishes (2 per cell line) at $2.0 \times 10^6$ cells per dish. The following day fusion protein expression was induced with 25 ng/mL doxycycline-containing medium for 24 h. For the BioID cell lines, biotin was also added at that point at a concentration of 50 µM. For the ultraID and TurboID cell lines, following the 24 h induction, the medium was replaced with 50 µM biotin and 25 ng/mL doxycycline-containing medium, and the cells were incubated at 37 °C for 10 min. PDB reactions were stopped by putting the plates on ice, removing the medium, and washing three times with ice-cold PBS. All further steps were performed at 4 °C. The cells were scraped in 1.5 mL PBS and collected in 15 mL tubes by centrifugation at $1200 \times g$ for 5 min at 4 °C. Cell pellets were frozen in liquid nitrogen and stored at −80 °C until further processing.

The procedure was similar for the P19 stable cell lines except that PDB was performed with three biological replicates for each sample (ultraID-Ago2, γ1-COP-ultraID, and γ2-COP-ultraID, with or without BFA addition) and fusion protein expression was induced with 5 ng/mL doxycycline for 24 h. After induction, the medium of the BFA-treated samples was exchanged for 5 µg/mL brefeldin A-containing medium, and the cells incubated at 37 °C for 1 h. Thereafter, the medium was exchanged for 5 µg/mL brefeldin A and 50 µM biotin-containing medium, and the cells were incubated for another 4 h. The mock-treated cells were incubated with 50 µM biotin-containing medium for 4 h. PDB was then stopped and sample storage was performed as for the HeLa cells.

**Streptavidin affinity pulldowns and sample preparation for MS analysis.** Cell pellets were thawed on ice and resuspended in 1 mL ice-cold RIPA buffer (50 mM Tris-HCl pH 8.0, 150 mM NaCl, 0.1% SDS, 0.5% sodium deoxycholate, 1% Triton X-100, 1x Protease Inhibitor Cocktail, 1 mM DTT). All subsequent steps were performed at 4 °C. 1 µL benzonase (Sigma) was added and the cell suspensions were transferred to RNase-free microcentrifuge tubes. Lysates were sonicated with a Bioruptor (Diagenode) at high intensity with 4 cycles of 30 s ON and 30 s OFF. The sonicated lysates were centrifuged at $16,000 \times g$ for 10 min. The cleared lysates were then transferred to a fresh microcentrifuge tube and 30 µL were stored as input material. Protein concentration was determined by a Bradford assay and all samples were adjusted to 2–3 mg protein in 1 mL RIPA buffer. High-performance streptavidin sepharose beads (Cytiva, #17-5113-01, Lot 10280314) were chemically modified to yield resistance against tryptic digestion as described in ref. [18]. 80 µL modified streptavidin-sepharose beads equilibrated in RIPA buffer were used per pulldown. The protein concentration-adjusted lysate samples were incubated with the equilibrated beads at 4 °C for at least 1 h. The beads were then harvested by centrifugation and washed with four consecutive buffers. For each wash, 1 mL of buffer was incubated with the beads at RT for 5 min, the beads were then harvested by centrifugation. The washing procedure included two washes with wash buffers 1, 2, and 3 each and three washes with wash buffer 4 (wash buffer 1: 2% SDS; wash buffer 2: 0.1% sodium deoxycholate, 1% Triton X-100, 500 mM NaCl, 1 mM EDTA, 50 mM HEPES pH 7.4; wash buffer 3: 10 mM Tris-HCl pH 8.0, 250 mM LiCl, 1 mM EDTA, 0.5% NP-40, 0.1% sodium deoxycholate; wash buffer 4: 50 mM ammonium bicarbonate). The buffer from the final wash was carefully removed and 1 µg of mass spectrometry grade trypsin (Serva, Heidelberg) in 60 µL wash

buffer 4 was added to each sample. On-beads digestion was performed at 37 °C overnight. On the following day, another 0.5 µg of trypsin was added to each sample and the digestion was resumed at 37 °C for another 2 h. The beads were centrifuged and the supernatant was carefully transferred to a fresh RNase-free microcentrifuge tube. The beads were washed with 30 µL of high-performance liquid chromatography (HPLC) grade water at RT for 5 min. After centrifugation, the supernatant was mixed with the previously collected supernatant and the elution procedure was repeated once. The eluates were centrifuged at $16'000 \times g$ for 5 min and 100 µL of the supernatant was transferred to a fresh microcentrifuge tube. The samples were acidified with 4 µL of 50% formic acid and dried in a SpeedVac centrifugal evaporator (ThermoFisher) at 65 °C for 2 h. The dried peptides were stored at −80 °C until analysis by mass spectrometry.

**Mass spectrometry data acquisition.** Dried peptide samples were reconstituted in 20 µL LC-MS sample buffer (2% acetonitrile, 0.1% formic acid). 2 µL of each sample were subjected to reverse-phase liquid chromatography for peptide separation using an RSLCnano Ultimate 3000 system (Thermo Fisher Scientific). Therefore, peptides were loaded on an Acclaim PepMap 100 pre-column (100 µm × 2 cm, C18, 5 µm, 100 Å; Thermo Fisher Scientific) with 0.07% trifluoroacetic acid at a flow rate of 20 µL/min for 3 min. Analytical separation of peptides was done on an Acclaim PepMap RSLC column (75 µm × 50 cm, C18, 2 µm, 100 Å; Thermo Fisher Scientific) at a flow rate of 300 nL/min. The solvent composition was gradually changed within 94 min from 96% solvent A (0.1% formic acid) and 4% solvent B (80% acetonitrile, 0.1% formic acid) to 10% solvent B within 2 min, to 30% solvent B within the next 58 min, to 45% solvent B within the following 22 min, and to 90% solvent B within the last 12 min of the gradient. All solvents and acids had Optima grade for LC-MS (Thermo Fisher Scientific). Eluting peptides were online ionized by nano-electrospray (nESI) using the Nanospray Flex Ion Source (Thermo Fisher Scientific) at 1.5 kV (liquid junction) and transferred to a Q Exactive HF mass spectrometer (Thermo Fisher Scientific). Full scans in a mass range of 300 to 1650 $m/z$ were recorded at a resolution of 30,000 followed by data-dependent top 10 HCD fragmentation at a resolution of 15,000 (dynamic exclusion enabled). LC-MS method programming and data acquisition were performed with the XCalibur 4.0 software (Thermo Fisher Scientific).

**Mass spectrometry data analysis.** The raw files from the MS analysis were analyzed with the MaxQuant software[19] for peptide and protein identification and assignment of LFQ intensities. The standard pre-assigned MaxQuant parameters were used with no activation of the "match between run" option (only peptides detected by MS/MS were considered). For enrichment analysis, the Maxquant data were analyzed in R using the packages proDA[21], EnhancedVolcano[61], and VennDiagram[62]. Proteins identified with proDA analysis were classified as significant hits if the $\log_2$ fold change was >2 and the $p$ value was <0.01. Few obvious hits were missed by the enrichment analysis by proDA and were added to the list of significantly enriched proteins. The criteria for enrichment for these hits were for the HeLa-11ht cell lines: detected in all four Ago2 samples with 2 unique peptides in at least three replicates, and no peptide detected in any Rab11 control sample. For the P19 cells: detected in all three γ1/2-COP samples with 2 unique peptides in all three replicates or 3 unique peptides in at least two replicates, and no peptide detected in any Ago2 control sample.

**Statistics and reproducibility.** The MS samples originated from 3 to 4 replicate pulldowns ($n = 3$ or $n = 4$). The proteins that were quantified in MaxQuant with ≥1 unique peptide and detected in at least one of the compared samples were selected for subsequent analysis. Data analyses were performed and results were visualized using R (Version 4.1.2). The LFQ values were imported from Maxquant using the import_MaxQuant() function (DEP, DOI: 10.18129/B9.bioc.DEP). For the LFQ-based quantification, a probabilistic dropout model was implemented using the proDA package with default parameters except for a maximum of 200 iterations. The test_diff() function (proDA package) was then used to perform a two-sided Wald test on two contrasted conditions. Hits with a fold change ≥2 and $p$ value <0.01 were deemed as significant. A typical and annotated R-script is provided in supplementary note 1.

**Library generation and yeast cultivation.** For randomization of microID, error-prone PCR was performed using the GeneMorph II Random Mutagenesis Kit (Agilent). Hereby, three individual PCRs with different mutation rates ranging from low to high were carried out. PCR products were purified using the *Promega Wizard SV Gel and PCR Clean-Up System* and amplified by conventional PCR using primers that incorporated 30 bp sequences homologous to the pCT display vector[63]. A yeast surface-display library was generated as described in refs. [14,64]. Gap repair cloning was performed into a *Bam*HI, *Xho*I, and *Nhe*I hydrolyzed pCT vector. Library size was estimated by plating a serial dilution of the regenerated yeasts on SD-Trp agar. The resulting library was cultivated in 1 L SD-Trp medium at 30 °C and 180 rpm. The next day, the surface presentation was induced by inoculating the library in 50 mL SG-Trp medium to an optical density of 1 and incubation at 30 °C and 180 rpm overnight.

**Yeast-surface biotinylation assay and staining**. In all, $1 \times 10^{7} - 1 \times 10^{8}$ yeasts were washed twice and resuspended in 1 mL PBS. Biotin and ATP were added to a final concentration of 50 μM and 2.5 mM, respectively. Cells were incubated at 30 °C and 900 rpm to prevent settling. Incubation time was decreased for each screening round. To prepare yeast cells for FACS screening, biotinylated cells were washed twice with PBS, resuspended in 100 μL PBS containing 0.1 μg anti-penta·His antibody (Qiagen) and 1 μg streptavidin allophycocyanin (Invitrogen) and incubated on ice for 20 min. After two washing steps, cells were stained as described before using 0.2 μg anti-mouse FITC antibody (Sigma-Aldrich). Finally, cells were washed again twice and were suspended in 1.5 mL of PBS.

**FACS screening**. FACS screening was performed after cell staining. Sorting gates were set in comparison to a microID negative control (no biotin and ATP). Cells exhibiting both, a fluorescence signal for biotin ligase activity (670 nm) and surface presentation (530 nm), were sorted using a BD Influx system and cultivated on SD-agar plates. After 2 days, cells were scraped off the plates, inoculated in SD liquid medium, and cultivated overnight. After induction of surface presentation, biotinylation and staining as described above, the next round of screening was performed.

**Recombinant expression**. microID, ultraID-4, and ultraID-5 genes were amplified by PCR and subcloned into a pET22b expression vector, which was used for the transformation of E. coli BL21 (DE3) cells. R40G/L41P and L41P-only mutants were generated by site-directed mutagenesis using the pET22b_microID plasmid as a template. A 50 mL dYT overnight culture was used to inoculate 1 L TB-medium to an $OD_{600} = 0.1$ and was incubated at 37 °C and 220 rpm. Enzyme expression was induced by the addition of 0.5 mM IPTG (final concentration) after reaching an $OD_{600}$ ~0.6. Expression was carried out overnight at 18 °C and 220 rpm. After cell lysis by sonication, the enzymes were purified by IMAC, dialyzed against 50 mM TRIS-HCl pH 7.5, 150 mM NaCl, and 5 mM $MgCl_2$ overnight, and stored at −80 °C.

**ELISA-based biotinylation assay**. To quantify enzymatic activity, a 96-well Nunc MaxiSorp plate (ThermoFisher scientific) was coated with 200 μL of a 40 mg/mL human serum albumin fraction V (CarlRoth, Germany) for 1 h. Wells were washed three times with PBST (6.4 mM $Na_2PO_4$, 2 mM $KH_2PO_4$, 10 mM KCl, 140 mM NaCl, 0,1% (w/v) Tween20, pH 7.4). For biotinylation reactions, 50 μM Biotin, 2.5 mM ATP, and 0.06 mg/ml biotin ligase (final concentration, each) in 50 mM TRIS-HCl pH 7.5, 150 mM NaCl, and 5 mM $MgCl_2$ were added and incubated for 10 min at 37 °C. The reaction was stopped by adding 0.25 mM EDTA. Subsequently, wells were washed three times with PBST. Alkaline phosphatase-conjugated streptavidin (Sigma-Aldrich) was diluted 1:5000 and added in a volume of 50 μL for 1 h at RT. Staining reaction was carried out after three washing steps and equilibration of the wells with AP-buffer. 1 mg/mL para-nitrophenyl phosphate was added and absorbance at 405 nm was measured with a TECAN Infinite® 200 PRO plate reader after 20 min of staining. Five individual assays were performed in triplicates to quadruplicates. After subtraction of the negative control (no BirA added), the mean microID absorbance was set to 100%.

**Reporting summary**. Further information on research design is available in the Nature Research Reporting Summary linked to this article.

## Data availability

The MS proteomics data have been deposited to the ProteomeXchange Consortium (http://proteomecentral.proteomexchange.org) via the PRIDE partner repository[65] with the data set identifiers PXD026719 (BioID-Rab11/Ago2 and ultraID-Rab11/Ago2 with 10 min labeling), PXD032979 (BioID/ultraID/TurboID-Ago2/Rab11 with no biotin addition + TurboID-Rab11/Ago2 with 10 min labeling) and PXD026715 (γ1/γ2-ultraID dataset). Raw data for the melting temperature and ELISA-based biotinylation assay are provided in Supplementary Data 4 and 5, respectively. Additional data that support the findings of this study are available from the corresponding author on request.

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

## Acknowledgements

We thank Valeria Mery, Jérôme Oswald, and Jannik Mattern for helping establish PDB in bacteria, Hans Dieter Schmitt (MPI for Multidisciplinary Sciences, Göttingen, Germany) for providing codon usage-optimized BioID2 for *S. cerevisiae* as a template for generating *ASC1-microID* and *ASC1-ultraID* plasmids. LC-MS analysis was done in the Service Unit LCMS Protein Analytics of the Göttingen Center for Molecular Biosciences (GZMB) at the Georg-August-University Göttingen (Grant DFG-GZ: INST 186/1230-1 FUGG to Stefanie Pöggeler). We are indebted to Mandy Jeske and Doris Höglinger (Heidelberg University) who provided laboratory space and support to L.K. during the revision time of this manuscript. Part of the work was financed by the excellence initiative of the German research council (DFG-EXC81 to J.B.).

## Author contributions

L.K. performed all experiments with HeLa cells and bacteria. S.B. and L.D. performed the directed evolution of microID. X.Z. performed all experiments with P19 cells. A.R. performed all the preliminary experiments leading to the discovery and initial characterization of microID. E.C.B. performed preliminary experiments for the characterization of microID and ultraID. K.S. and O.V. contributed to the yeast experiments and the LC-MS analysis. H.K. supervised the directed evolution approach. J.B. designed and supervised the study, and analyzed the data. L.K. and J.B. wrote the first draft of the manuscript. All authors edited the paper.

## Funding

## Competing interests

The authors declare no competing interests.
