## [Peer Review File · Communications Biology]

Reviewers' comments:

Reviewer #1 (Remarks to the Author):

Brief summary:

Proximity biotinylation is an orthogonal approach to affinity purification for interactome mapping experiments. It is especially suited for the analysis for hard to solubilize or for membrane embedded bait proteins which are refractory to traditional approaches. With this in mind, Zhao et al. were interested in generating a faster version of their previously disclose split-BioID system. To do so, they employed the published structure of BioID2 biotin ligase and designed a cut-site to create two independent fragments NBioID2

79 (BioID2 [2-171]) and CBioID2 (BioID2 [172-233]). Surprisingly, the authors found that the NBioID2 fragment remained active on its own making it the smallest abortive biotin ligase to date. They rename this enzyme microID. microID was shown by western blot to be highly effective with an overall biotinylation activity in time-frame as little as 10 minutes. Next, they improve the activity of microID through protein engineering in yeast to generate a novel enzyme termed ultraID. Biophysical characterization of ultraID showed that two mutations (R40G and L41P) were sufficient to explain the gain in activity between microID and ultraID. ultraID was shown to be effective in mammalian cells, yeast and bacteria by the authors. ultraID was shown to be effective in proximity biotinylation experiments coupled to mass spectrometry for the Ago2, Rab11, γ 1-COP and γ 2-COP proteins.

Overall impression of the work:

The manuscript by Zhao et al. is of high quality, well prepared and easy to read. microID and ultraID are important reagents that will be of high interest to a wide audience for future interactome mapping experiments.

Specific comments:

- A direct comparison between ultraID and TurboID should be provided to strengthen some of the claims in the manuscript. The current Figure 7 in which ultraID is contrasted to BioID is not ideal to truly highlight the benefit of ultraID for fast interactome mapping experiments.

- On lines 356-359, the authors refer to a BioID-TurboID comparison without references or data. This should be clarified to strengthen the arguments put forward by the authors.

- The authors should try to showcase better the biggest benefits of their microID and ultraID enzymes, namely their smaller size and lack of DNA binding domains. To do so, the authors may want to show how ultraID fusion with large proteins remain more effective than the TurboID. Similarly, they may want to show for nuclear proteins whether ultraID tag is more specific than TurboID to assess whether the lack of DNA binding domains affect interactome mapping experiments.

- The parameters employed to perform the proDA statistical analysis of ultraID MS data should be clarified to help readers repeat these analyses themselves.

- Input controls for many western blots (Figure 1, 2, 4, 5, 6B and 8) are missing. While the results themselves are of high quality, the lack of input controls makes the comparison between the streptavidin signal more difficult. Please provide these blots to ensure the validity of the conclusions derived from these experiments.

- On line 807, the appropriate information regarding the data deposition is missing. Please ensure MS data deposition and disclosure of the proper information.

- What is the optimal temperature to use ultraID? Since it was engineered in yeast (30°C), it may be more active at 30°C than 37°C as is the case for TurboID.

Reviewer #2 (Remarks to the Author):

This manuscript by Zhao et al., describes the generation and application of a new BioID variant called UltraID. UltraID was compared to most of the existing BioID ligase variants and appears to be smaller than all, and comparable to the more recent TurboID in terms of labeling efficiency. There is also the claim that UltraID generates less background biotinylation than TurboID, although this is not convincing as presented, but might be the case with further studies suggested below. Finally, the new ligase was used to probe associations of Ago2 and a component of the COPI complex.

Most if not all Western blots lack loading controls. This can be important in interpreting the ligase efficacy as it impacts the abundance of background labeling, including endogenously biotinylated proteins, as well as interpreting relative expression levels of the ligases themselves.

The data showing that the various ligases require less biotin are fine in principle, but the concept is potentially flawed without further data. Less biotin than what, especially when there is no control for expression levels? You could do the same experiment with any of the ligases and see a diminishing level of biotinylation. The point is when does it reach saturation, the point at which more biotin is not useful and/or what is the least needed to be practically useful which probably would require MS results and be situationally dependent. I am not suggesting the latter, but the former point of when it reaches saturation might be useful and experimentally comparing it to another ligase would be necessary to make the claim that it requires less biotin. This kind of comparison was done in fig 2 of PMID: 26912792 and shows similar results.

There seems to be an increase in basal biotinylation (occurring in the absence of biotin supplementation) that correlated with the activity of the ligase. While this is not surprising, it does come at the risk that identified candidate might have been labeled prior to the supplementation period. Based on the data provided I am not yet convinced that there is less background labeling than TurboID under basal conditions. If this claim is to be made with fully supportive data I would suggest a practical test of reduced background would be to do a parallel comparative pulldown/MS analysis of a TurboID vs UltraID for a bait, with proper controls, at time 0 (no biotin supplementation versus 10 min biotin supplementation) and preferably try to have the levels of the ligase and lysate at least relatively similar between the ligases. If you detect the same proteins (typically the more abundant ones) at time 0 as you do at 10 min for either ligase then the ligase cannot be accurately used to assess association that happened specifically during the labeling period (as was seen in fig 7 and tables S5-6 PMID: 32344865). And if UltraID detects fewer of those proteins at time 0 than does TurboID then it indeed does have less background than TurboID. These BioID variants that require less biotin, BioID2 included, do run the risk of enhanced biotinylation under basal conditions especially since some media formulations include biotin (like RMPI 1640) leading to constant biotinylation, and even some batches of serum seem to have more biotin than others and can cause unwanted background biotinylation. The use of inducible expression is an approach that can help reduce but on its own will not eliminate this background labeling. Even BioID2, which UltraID is derived from, benefitted from biotin depletion in considerably reducing this background.

The addition of dox and biotin simultaneously for BioID experiments is fine, but the labeling is probably not accurately 24 hrs as it takes time for the dox to generate protein expression. I am not suggesting that this considerably changes your results since 15-18 hrs is typically enough to 'saturate' biotinylation for BioID, but you might think of a way to highlight this technical discrepancy in the presentation of your results.

Reviewer #3 (Remarks to the Author):

This is a nice study in which the authors engineer a novel version of a biotin protein ligase (BPL) for proximity dependent biotinylation (PDB) and mass spectrometry experiments. Starting with the BPL BioID2, and guided by their own previous work generating a split BioID enzyme, they first generate an NT fragment that retains catalytic activity (microID), followed by employing a directed evolution strategy to introduce mutations to enhance the activity further (ultraID). They benchmark their enzymes against existing BPLs, and demonstrate their utility in a few different

settings (yeast, bacteria, and mammalian cells). The authors claim that the resulting novel enzymes offer distinct advantages over existing tools, including a smaller size and low levels of background labelling coupled with rapid kinetics. Both microID and ultraID will be valuable tools in the ever-expanding BPL toolkit. I have the following questions and points of consideration for the authors.

1. It's intriguing that activity was retained in the NT fragment of BioID2 (i.e., microID) after making a split at the site analogous to the one the authors had previously made in BirA*. Was any degree of activity retained in the NT fragment of BirA* when making the split in that case? If not, is there any predicted structural explanation for why activity may have been retained in the case of microID but not in BirA*? It might be helpful to indicate the location of R118G/R40G and/or the catalytic pocket in the structures shown in Figure 1A.

2. The authors claim that ultraID displays "markedly less" background labeling than TurboID. Based on their experimental results, this claim seems a bit exaggerated, and the authors might benefit from softening their stance on this throughout the paper. Although background may be slightly reduced with ultraID, it seems to track somewhat linearly with overall activity (for example in Figure 5). Is background labeling not simply a function of enzyme activity level? Or is there some other feature of the enzyme that could account for reduced background that is not purely tied to activity level? If the authors do indeed believe there is reduced background inherent to ultraID that is not explained by activity level, then perhaps this should receive a comment in the paper.

3. When testing micro/ultraID fused to bait proteins (yeast/bacteria/mammalian cells) why is the comparison made only with BirA*/BioID? Wouldn't a direct comparison with TurboID make sense here (as was done in previous experiments looking at non-fused enzymes). A major claim of the paper is that ultraID has similar activity but lower background compared to TurboID, so it strikes me that direct comparison should be made between the two in this experimental setup as well.

4. The authors suggest that due to the smaller size of micro/ultraID it should have less of a negative impact on function/trafficking of bait proteins. Is this purely a theoretical claim at this point, or do they have experimental evidence that this may be the case? If so, demonstrating this would really strengthen their case that the smaller enzyme size is likely to offer a practical benefit.

5. It seems like an odd choice to use Rab11 as a control against AGO2. Can the authors please provide a rationale for why this control was chosen? Part of why this seems like an odd choice is that AGO2 has been reported to localize to populations of endosomes (e.g., PMIDs: 19684575, 24723684) therefore there may be shared proximity interactions between Rab11 and AGO2 through localizing to shared structures. Even if Rab11 is used as a control, an enzyme-only control (i.e., BioID or ultraID not fused to a bait) would also be valuable for filtering mass spectrometry data.

6. Can the authors please comment on the subset of preys that were specific to ultraID and/or BioID in their AGO2/Rab11 comparison (i.e., right and left of Venn diagram in Fig 7B)? Is there something specific about these two sets of unique preys that might suggest bias in the type of compartments/preys that these two enzymes are able to label?

7. Please indicate the cell type that was used for the experiment shown in Figure 7, either in the main text or the figure legend. Also, I'm not sure that "authentic experimental conditions" is the best wording to use in the section title.

8. In the coatomer PDB-MS experiment, I again think that an enzyme-only control could be included to filter mass spectrometry data. And here too, the rationale for using AGO2 as a control against coatomer subunits is not entirely clear to me.

9. In Figure 8 why is Arf not followed by a number? What gene is this?

10. In my opinion, the section describing the coatomer PDB-MS results is too lengthy. This section

would benefit from being condensed and/or moving some of this information to the Discussion section.

Dear reviewers,

We would first like to thank you for your insightful and constructive comments. Given the unusual circumstances over last year (including the dissolution of my group in Heidelberg) we have made our best to address the main raised issues in our manuscript.

Please note that while revising the manuscript we noticed a copy/paste mistake from an Excel table that affected the calculation of the activities of the enzymes in the ELISA-based assay (Fig. 3E). Fortunately the incorrect values were marginally different from the correct ones, which are implemented in the revised manuscript.

Below, you will find a point-by-point answer to your comments. As some time has elapsed between the initial submission and the current re-submission, we provide a side-by-side comparison of the original and revised text when relevant. We hope that this will be helpful for you. Here and in the accompanying revised manuscript, changes are highlighted in yellow.

Reviewer #1 (Remarks to the Author):

Brief summary:

Proximity biotinylation is an orthogonal approach to affinity purification for interactome mapping experiments. It is especially suited for the analysis of hard to solubilize or for membrane embedded bait proteins which are refractory to traditional approaches. With this in mind, Zhao et al. were interested in generating a faster version of their previously disclosed split-BioID system. To do so, they employed the published structure of BioID2 biotin ligase and designed a cut-site to create two independent fragments NBioID2 79 (BioID2 [2-171]) and CBioID2 (BioID2 [172-233]). Surprisingly, the authors found that the NBioID2 fragment remained active on its own making it the smallest abortive biotin ligase to date. They rename this enzyme microID. microID was shown by western blot to be highly effective with an overall biotinylation activity in time-frame as little as 10 minutes. Next, they improve the activity of microID through protein engineering in yeast to generate a novel enzyme termed ultraID. Biophysical characterization of ultraID showed that two mutations (R40G and L41P) were sufficient to explain the gain in activity between microID and ultraID. ultraID was shown to be effective in mammalian cells, yeast and bacteria by the authors. ultraID was shown to be effective in proximity biotinylation experiments coupled to mass spectrometry for the Ago2, Rab11, γ 1-COP and γ 2-COP proteins.

Overall impression of the work:

The manuscript by Zhao et al. is of high quality, well prepared and easy to read. microID and ultraID are important reagents that will be of high interest to a wide audience for future interactome mapping experiments.

Thank you very much for your positive assessment.

Specific comments:

1. A direct comparison between ultraID and TurboID should be provided to strengthen some of the claims in the manuscript. The current Figure 7 in which ultraID is contrasted to BioID is not ideal to truly highlight the benefit of ultraID for fast interactome mapping experiments.

We have added a comparison to TurboID using two new stable cell lines for TurboID-Ago2 and TurboID-Rab11. Figure 7 has been updated accordingly. Please note that we prepared a new batch of chemically-modified streptavidin beads for the pulldowns performed for the revision work. This may explain the unexpected relatively low number of hits we obtained with TurboID. We comment this in the revised manuscript.

2. On lines 356-359, the authors refer to a BioID-TurboID comparison without references or data. This should be clarified to strengthen the arguments put forward by the authors.

Since we now directly compare ultraID, TurboID and BioID we have removed this sentence from the revised manuscript and comment our own data. The study referred to is mentioned in the discussion with an appropriate reference (lanes 517-518 as in original manuscript).

3. The authors should try to showcase better the biggest benefits of their microID and ultraID enzymes, namely their smaller size and lack of DNA binding domains. To do so, the authors may want to show how ultraID fusion with large proteins remain more effective than the TurboID. Similarly, they may want to show for nuclear proteins whether ultraID tag is more specific than TurboID to assess whether the lack of DNA binding domains affect interactome mapping experiments.

A previous report (Kim et al., 2014) made a direct comparison of fusions of either BioID (same size as TurboID) or BioID2 (bigger than ultraID) to the inner nuclear membrane protein Sun2. It demonstrated that BioID was bulky enough to prevent the correct localization of the fusion protein to the INM whereas the BioID2-Sun2 fusion showed a correct localization. Since UltraID is even smaller than BioID2 and the localization of Sun2 fusion proteins is directly dependent on the size of the fusion partner (Ungricht, R. et al. JCB 2015, our ref. 54), we refer to this study to indicate that a smaller labeling enzyme is an advantage in some cases (“To date, ultraID is by far the smallest enzyme for PDB, 44% smaller than TurboID/BioID and 25% smaller than BioID2 (19.7 vs. 35 and 26.4 kDa respectively). This **may** also be advantageous as it is generally considered that the larger the size of a tag, the greater the chance of interfering with the function, trafficking and interactions of the fusion protein⁵³. For example, BioID, which has the same size as TurboID, prevents the correct localization of the protein Sun2 to the INM⁴, a process that is sensitive to the size of the fusion partner⁵⁴. By contrast, BioID2, which is larger than ultraID, leads to a correct INM localization when fused to Sun2⁴.”, lanes 508-515).

We had no time and available resources to test interactome of nuclear proteins. We would still like to keep the comment in the introduction on the potential risk of having a DNA-binding domain as it may lead to non-specific binding to the host DNA when the fusion protein is expressed in the nucleus. We tried to formulate it not as a fact but as a word of caution. However we are ready to remove this comment if this is the reviewer wish.

Original text:

Finally, as *E. coli*'s BirA is a class II biotin ligase, it has an N-terminal DNA-binding domain⁵ that shows structural homology to linker histone H5⁶ and may lead to artefactual non-specific binding to the host DNA and/or chromatin-interacting proteins.

Revised text (lines 37-41):

Finally, as *E. coli*'s BirA is a class II biotin ligase, it has an N-terminal DNA-binding domain⁵ that shows structural homology to linker histone H5⁶ and **that theoretically might** lead to artefactual non-specific binding to the host DNA and/or chromatin-interacting proteins, **though this has not been investigated so far.**

4. The parameters employed to perform the proDA statistical analysis of ultraID MS data should be clarified to help readers repeat these analyses themselves.

More details are now provided in the method section in the paragraph “statistics and reproducibility” (lines 754-765). We now also provide a typical and annotated R script we used for the MS data analysis so that the reader can repeat the analyses in the exact same way they were performed (Supplementary note 1).

5. Input controls for many western blots (Figure 1, 2, 4, 5, 6B and 8) are missing. While the results themselves are of high quality, the lack of input controls makes the comparison between the streptavidin signal more difficult. Please provide these blots to ensure the validity of the conclusions derived from these experiments.

We usually use the signal from the endogenous biotinylated proteins as an internal loading control as this is more reliable than using yet another detection reagent. We acknowledge however that endogenous biotinylated proteins are not clearly visible in the samples with high biotinylation activity at 1h and overnight labeling time. For some of the blots mentioned by the reviewers we could use a cross-reactivity signal observed with the myc antibody as internal loading control, otherwise, we have re-performed experiments for the indicated figures and added a loading control. The figures have been updated accordingly. Note that for Fig. 8 we had already probed the membranes with an anti-GAPDH antibody, so we just added the data to the original figure.

6. On line 807, the appropriate information regarding the data deposition is missing. Please ensure MS data deposition and disclosure of the proper information.

The MS data are available through the PRIDE repository, accession numbers are indicated in the revised manuscript (lines 830-837).

- What is the optimal temperature to use ultraID? Since it was engineered in yeast (30°C), it may be more active at 30°C than 37°C as is the case for TurboID.

We do not know the answer to this question at the moment. UltraID is thermostable as in preliminary experiments we observed strong activity in *Chaetomium thermophilum* (growth at 50°C). In *E. coli* we observed stronger activity at 37°C than 30°C but the whole metabolism of the cells is of course also affected by the change of temperature, making direct conclusions on the optimal temperature for the enzyme difficult. Same goes for comparison between organisms. At the moment we only claim that the enzyme can be used in yeast at 30°C or bacteria/mammalian cells at 37°C. Other labs are currently testing the enzyme in other organisms growing at various temperatures so we expect more experimental data will be available in the near future.

Reviewer #2 (Remarks to the Author):

This manuscript by Zhao et al., describes the generation and application of a new BioID variant called UltraID. UltraID was compared to most of the existing BioID ligase variants and appears to be smaller than all, and comparable to the more recent TurboID in terms of labeling efficiency. There is also the claim that UltraID generates less background biotinylation than TurboID, although this is not convincing as presented, but might be the case with further studies suggested below. Finally, the new ligase was used to probe associations of Ago2 and a component of the COPI complex.

1. Most if not all Western blots lack loading controls. This can be important in interpreting the ligase efficacy as it impacts the abundance of background labeling, including endogenously biotinylated proteins, as well as interpreting relative expression levels of the ligases themselves.

Please see answer to comment #5 from Reviewer #1

2. The data showing that the various ligases require less biotin are fine in principle, but the concept is potentially flawed without further data. Less biotin than what, especially when there is no control for expression levels? You could do the same experiment with any of the ligases and see a diminishing level of biotinylation. The point is when does it reach saturation, the point at which more biotin is not useful and/or what is the least needed to be

practically useful which probably would require MS results and be situationally dependent. I am not suggesting the latter, but the former point of when it reaches saturation might be useful and experimentally comparing it to another ligase would be necessary to make the claim that it requires less biotin. This kind of comparison was done in fig 2 of PMID: 26912792 and shows similar results.

Thank you for the comment and point well taken. We have removed the data from the manuscript and agree they were per se not very informative.

3. There seems to be an increase in basal biotinylation (occurring in the absence of biotin supplementation) that correlated with the activity of the ligase. While this is not surprising, it does come at the risk that identified candidate might have been labeled prior to the supplementation period. Based on the data provided I am not yet convinced that there is less background labeling than TurboID under basal conditions. If this claim is to be made with fully supportive data I would suggest a practical test of reduced background would be to do a parallel comparative pulldown/MS analysis of a TurboID vs UltraID for a bait, with proper controls, at time 0 (no biotin supplementation versus 10 min biotin supplementation) and preferably try to have the levels of the ligase and lysate at least relatively similar between the ligases. If you detect the same proteins (typically the more abundant ones) at time 0 as you do at 10 min for either ligase then the ligase cannot be accurately used to assess association that happened specifically during the labeling period (as was seen in fig 7 and tables S5-6 PMID: 32344865). And if UltraID detects fewer of those proteins at time 0 than does TurboID then it indeed does have less background than TurboID. These BioID variants that require less biotin, BioID2 included, do run the risk of enhanced biotinylation under basal conditions especially since some media formulations include biotin (like RPMI 1640) leading to constant biotinylation, and even some batches of serum seem to have more biotin than others and can cause unwanted background biotinylation. The use of inducible expression is an approach that can help reduce but on its own will not eliminate this background labeling. Even BioID2, which UltraID is derived from, benefitted from biotin depletion in considerably reducing this background.

Thank you for this comment and the suggestion of a suitable experiment that we tried to follow. We have used TurboID and ultraID cell lines and performed pulldowns at time 0. With our HeLa cell lines grown in DMEM we did not detect many hits for both sets of cell lines. We quantified the background by comparing the abundance (iBAQ) of endogenous biotinylated proteins, fusion proteins and two core interacting partners (TNRC6A & B for the Ago2 cell lines, and RABFIP1 & 5 for Rab11 cell lines) that are typically the most abundant hits in the Ago2 and Rab11 PDB-MS datasets. This is the new panel C in revised Fig. 7. With this experiment, we observed a lower abundance of fusion proteins and core interacting partners from the ultraID cell lines. In line with the first part of your comment, and comment #2 of reviewer #3, we describe the observation as a "more favorable balance signal vs background for ultraID".

This corresponds to lanes 367-396 in the revised manuscript:

“To compare the background labeling activity of ultraID and TurboID prior to biotin addition, we also performed streptavidin pulldowns with the same batch of beads from cells that were not incubated with extra biotin. In the corresponding MS datasets, the number of hits with consistent detection in all three TurboID/ultraID-Ago2 replicates was small (respectively 14 and 13 hits with at least two assigned LFQ values across replicates) and the proDA analysis did not converge. To assess the labeling background, we thus analyzed the abundance of the fusion proteins and two core interacting partners in each sample using the iBAQ (intensity Based Absolute Quantification) algorithm²⁷. The abundance of Ago2 and its two interacting partners TNRC6A and TNRC6B²⁸ was lower (at least 15 fold) in the pulldown from the ultraID-Ago2 cell line than from the TurboID-Ago2 cell line. Consistently the abundance of Rab11 was 15 fold lower in the pulldown from the ultraID-Rab11 cell line than from the TurboID-Rab11 cell line and the two core interacting partners of Rab11, RAB11FIP1 and RAB11FIP5¹⁷, were not detected at all in the ultraID-Rab11 cell line (Fig. 7C and supplementary table 2).

Together, our data show that ultraID allows obtaining relevant proteomic datasets in PDB experiments with a labeling time of 10 min at physiological expression levels and may provide a favorable balance of signal vs. background when compared to TurboID.”

And revised figure 7C:

4. The addition of dox and biotin simultaneously for BioID experiments is fine, but the labeling is probably not accurately 24 hrs as it takes time for the dox to generate protein expression. I am not suggesting that this considerably changes your results since 15-18 hrs is typically enough to 'saturate' biotinylation for BioID, but you might think of a way to highlight this technical discrepancy in the presentation of your results.

Thanks for this comment, it is true that it is not strictly speaking 24h. It is actually difficult to give a precise number: we know that with this expression system protein expression is seen about 1h after addition of dox (Béthune et al., 2012) but at that time the protein levels are so small that labeling is probably not significant. As a pragmatic solution we now write overnight labeling and define overnight as addition of dox and biotin 24h before cell lysis.

Reviewer #3 (Remarks to the Author):

This is a nice study in which the authors engineer a novel version of a biotin protein ligase (BPL) for proximity dependent biotinylation (PDB) and mass spectrometry experiments. Starting with the BPL BioID2, and guided by their own previous work generating a split BioID enzyme, they first generate an NT fragment that retains catalytic activity (microID), followed by employing a directed evolution strategy to introduce mutations to enhance the activity further (ultraID). They benchmark their enzymes against existing BPLs, and demonstrate their utility in a few different settings (yeast, bacteria, and mammalian cells). The authors claim that the resulting novel enzymes offer distinct advantages over existing tools, including a smaller size and low levels of background labelling coupled with rapid kinetics. Both microID and ultraID will be valuable tools in the ever-expanding BPL toolkit.

Thank you very much for your encouraging assessment.

I have the following questions and points of consideration for the authors.

1. It's intriguing that activity was retained in the NT fragment of BioID2 (i.e., microID) after making a split at the site analogous to the one the authors had previously made in BirA*. Was any degree of activity retained in the NT fragment of BirA* when making the split in that case? If not, is there any predicted structural explanation for why activity may have been retained in the case of microID but not in BirA*? It might be helpful to indicate the location of R118G/R40G and/or the catalytic pocket in the structures shown in Figure 1A.

Thanks for this comment, this is a question that also intrigued us. When splitting BioID we hardly saw any activity for the NT fragment (see for example Schopp et al., 2017, Fig. 5) and there is now obvious structural explanation that we can provide. Our guess is that coming from an extremophile, the structure of BioID2 may be more rigid and that might make the NT domain less prone to destabilization when the CT domain is removed. But this is just a guess that we have included in the revised discussion ("As removal of the C-terminal domain of BioID yields a fragment with

hardly any detectable activity¹², it was not expected that microID would be even more efficient than BioID2. A possible explanation may be the origin of BioID2. Coming from an extremophile organism, its overall structure is likely more rigid than that of BioID, possibly making its core catalytical domain less prone to destabilization when the C-terminal domain is removed.”, lanes 553-558).

For the second part of the comment: we have indicated the R118 and R40 on the structures shown in Fig. 1A as suggested. Note that we now use two other PDB files as in the original ones as the residues involved in biotin binding were not visible.

2. The authors claim that ultraID displays “markedly less” background labeling than TurboID. Based on their experimental results, this claim seems a bit exaggerated, and the authors might benefit from softening their stance on this throughout the paper. Although background may be slightly reduced with ultraID, it seems to track somewhat linearly with overall activity (for example in Figure 5). Is background labeling not simply a function of enzyme activity level? Or is there some other feature of the enzyme that could account for reduced background that is not purely tied to activity level? If the authors do indeed believe there is reduced background inherent to ultraID that is not explained by activity level, then perhaps this should receive a comment in the paper.

Thanks for this comment and point well taken: we have tried to reformulate the comments about background activity so that they are not qualitative and stick to the data. Also as suggested by Reviewer #2, comment # 3, we tried to measure by MS the labeling background generated by TurboID and ultraID fusion proteins when no extra biotin is added.

Original text:

Together, our data show that ultraID yields a labeling efficiency similar to TurboID at labeling times down to 10 min, with the advantage of a lower background biotinylation activity, making it a more efficient enzyme for PDB

Revised text (lines 285-287):

Together, our data show that ultraID yields a labeling efficiency similar to TurboID at labeling times down to 10 min, with the **potential** advantage of a lower background biotinylation activity, making it a more efficient enzyme for PDB.

Original text:

A side-by-side comparison of BioID and TurboID in which both enzymes were fused to the same specific bait proteins revealed that a 10 min labeling TurboID-derived dataset is larger than an 18 h labeling BioID-derived dataset but that the additional hits are mainly non-relevant proteins. This is probably due to a permanent background labeling activity of TurboID leading to a much wider labeling radius¹⁰. Our data suggest that ultraID is not affected by this shortcoming. Indeed, the ultraID-derived Ago2 interactome obtained after 10 min labeling was somewhat smaller (50 vs 68 hits) than that obtained from BioID after 24 h labeling (Figure 7B) but ultraID performed equally well to BioID in identifying known Ago2-associated proteins such as TNRC6 proteins²¹, Dicer²², CNOT1^{23, 24}, GIGYF

proteins¹² or XRN1²⁵ (Figure 7B, C) and the volcano plots resulting from both datasets were similar (Figure 7C).

Hence, ultraID allows obtaining relevant proteomic datasets in PDB experiments with a labeling time of 10 min at physiological expression levels.

Revised text (lines 354-396):

The ultraID-derived Ago2 interactome obtained after 10 min labeling was somewhat smaller (50 vs 68 hits) than that obtained from BioID after overnight labeling (Figure 7B) but ultraID performed equally well to BioID in identifying known Ago2-associated proteins such as TNRC6 proteins²², Dicer²³, CNOT1^{24, 25}, GIGYF proteins¹² or XRN1²⁶ (Figure 7B) and the volcano plots resulting from both datasets were similar (**Supplementary Fig. 10**). We obtained a smaller number of hits (17) from the TurboID dataset but they also represented relevant proteins (Figure 7B). This small number was not expected but may be due to the use of another batch of streptavidin beads for this series of pulldowns. The subset of preys specific to the BioID, ultraID or TurboID datasets also contains relevant proteins, such as CNOT11, DCP1A/B, PABPC1/4 for BioID; FXR1/2, PUM1 for ultraID and CNOT8 for TurboID (Supplementary Data 1 and 2). Hence a labeling bias towards a certain type of protein and/or localization is not obvious from our data.

To compare the background labeling activity of ultraID and TurboID prior to biotin addition, we also performed streptavidin pulldowns with the same batch of beads from cells that were not incubated with extra biotin. In the corresponding MS datasets, the number of hits with consistent detection in all three TurboID/ultraID-Ago2 replicates was small (respectively 14 and 13 hits with at least two assigned LFQ values across replicates) and the proDA analysis did not converge. To assess the labeling background, we thus analyzed the abundance of the fusion proteins and two core interacting partners in each sample using the iBAQ (intensity Based Absolute Quantification) algorithm²⁷. The abundance of Ago2 and its two interacting partners TNRC6A and TNRC6B²⁸ was lower (at least 15 fold) in the pulldown from the ultraID-Ago2 cell line than from the TurboID-Ago2 cell line. Consistently the abundance of Rab11 was 15 fold lower in the pulldown from the ultraID-Rab11 cell line than from the TurboID-Rab11 cell line and the two core interacting partners of Rab11, RAB11FIP1 and RAB11FIP5¹⁷, were not detected at all in the ultraID-Rab11 cell line (Fig. 7C).

Together, our data show that ultraID allows obtaining relevant proteomic datasets in PDB experiments with a labeling time of 10 min at physiological expression levels and may provide a favorable balance of signal vs. background when compared to TurboID.

3. When testing micro/ultraID fused to bait proteins (yeast/bacteria/mammalian cells) why is the comparison made only with BirA*/BioID? Wouldn't a direct comparison with TurboID make sense here (as was done in previous experiments looking at non-fused enzymes). A major claim of the paper is that ultraID has similar activity but lower background compared to TurboID, so it strikes me that direct comparison should be made between the two in this experimental setup as well.

Thanks for this comment that was also a request from Reviewer #1. We have added a comparison with TurboID in the revised manuscript (see answer to Comment #1 of Reviewer #1)

4. The authors suggest that due to the smaller size of micro/ultraID it should have less of a negative impact on function/trafficking of bait proteins. Is this purely a theoretical claim at this point, or do they have experimental evidence that this may be the case? If so, demonstrating this would really strengthen their case that the smaller enzyme size is likely to offer a practical benefit.

Reviewer #1 had a similar comment, please see answer to his comment #3

5. It seems like an odd choice to use Rab11 as a control against AGO2. Can the authors please provide a rationale for why this control was chosen? Part of why this seems like an odd choice is that AGO2 has been reported to localize to populations of endosomes (e.g., PMIDs: 19684575, 24723684) therefore there may be shared proximity interactions between Rab11 and AGO2 through localizing to shared structures. Even if Rab11 is used as a control, an enzyme-only control (i.e., BioID or ultraID not fused to a bait) would also be valuable for filtering mass spectrometry data.

Thanks for this comment. As a negative control we usually like to pick a protein with unrelated function but with a somewhat similar localization to the POI. That was the reason to pick Rab11 as a negative control as it fulfills these criteria. We acknowledge that proximal proteins to both Ago2 and Rab11 may be filtered out but we believe that Ago2 and Rab11 are so unrelated that this is unlikely. We are not convinced adding an enzyme-only control would improve the filtering of the MS data significantly: filtering of random/enzyme-specific labeling is already performed by using the data set obtained with the Rab11-enzyme fusion protein.

We acknowledge that the (important) discussion about the best negative control is probably not closed with our answer, and would deserve more (probably in person) discussion. Yet the goal here was to compare different enzymes by PDB-MS applying the same conditions. As we applied the same negative control for all three enzymes, we believe that even if one may argue that another negative control might have led to more precise datasets, the strategy is sufficient to address the comparison we wanted to make.

6. Can the authors please comment on the subset of preys that were specific to ultraID and/or BioID in their AGO2/Rab11 comparison (i.e., right and left of Venn diagram in Fig 7B)? Is there something specific about these two sets of unique preys that might suggest bias in the type of compartments/preys that these two enzymes are able to label?

We have added a comment in the main text ("The subset of preys specific to the BioID, ultraID or TurboID datasets also contains relevant proteins, such as CNOT11, DCP1A/B, PABPC1/4 for BioID; FXR1/2, PUM1 for ultraID and CNOT8 for TurboID (Supplementary Data 1 and 2). Hence a labeling bias towards a certain type of

protein and/or localization is not obvious from our data.”, lanes 362-366). The hits and subsets of hits that are specific to one enzyme are indicated in Supplemental Data 1.

7. Please indicate the cell type that was used for the experiment shown in Figure 7, either in the main text or the figure legend. Also, I’m not sure that “authentic experimental conditions” is the best wording to use in the section title.

Thanks for the hint, we added the cell type (HeLa) and changed the title of this section

8. In the coatomer PDB-MS experiment, I again think that an enzyme-only control could be included to filter mass spectrometry data. And here too, the rationale for using AGO2 as a control against coatomer subunits is not entirely clear to me.

For the coatomer PDB-MS, we chose Ago2 as a negative control because it is functionally non-related to coatomer but has a partial localization at the Golgi/ER interface like coatomer (the original name of human Ago2 was actually GERp95 for Golgi/ER protein of 95 kDa). With these properties we think Ago2 is probably is very good negative control to filter non-relevant hits. We have added a sentence to explain the choice of Ago2 as a negative control.

9. In Figure 8 why is Arf not followed by a number? What gene is this?

Thanks for pointing this typo: the Arf1,2,3,4 & 5 paralogs have almost identical protein sequences so the identification was ambiguous. We now indicate all the possible paralogs.

10. In my opinion, the section describing the coatomer PDB-MS results is too lengthy. This section would benefit from being condensed and/or moving some of this information to the Discussion section.

We have shortened this section; we would also be ready to move it to a supplementary note.

REVIEWERS' COMMENTS:

Reviewer #1 (Remarks to the Author):

I wish to congratulate the authors for excellent revisions to their manuscript. They have now addressed the majority of my concerns successfully. I am convinced that the new UltraID and microID reagents reported here will further numerous studies employing proximity dependent biotinylation assays. I fully support the publication of the revised manuscript.

Reviewer #2 (Remarks to the Author):

The additional experiments and revisions to the manuscript address all of my initial critiques and I do not have an additional ones. This tool should prove useful as an alternative method for proximity labeling. I eagerly await the opportunity to test it out.

Reviewer #3 (Remarks to the Author):

The authors have addressed all of my concerns. Congratulations on a very nice paper.

REVIEWERS' COMMENTS:

Reviewer #1 (Remarks to the Author):

I wish to congratulate the authors for excellent revisions to their manuscript. They have now addressed the majority of my concerns successfully. I am convinced that the new UltraID and microID reagents reported here will further numerous studies employing proximity dependent biotinylation assays. I fully support the publication of the revised manuscript.

Reviewer #2 (Remarks to the Author):

The additional experiments and revisions to the manuscript address all of my initial critiques and I do not have an additional ones. This tool should prove useful as an alternative method for proximity labeling. I eagerly await the opportunity to test it out.

Reviewer #3 (Remarks to the Author):

The authors have addressed all of my concerns. Congratulations on a very nice paper.

We thank the reviewers for their insightful comments that allowed us to improve our manuscript.